# Diversifying Parallel Ergodic Search:
# A Signature Kernel Evolution Strategy

**Sreevardhan Sirigiri**
The University of Sydney, Australia
ssir4919@uni.sydney.edu.au

**Christian Hughes**
Yale University, USA
christian.hughes@yale.edu

**Ian Abraham**
Yale University, USA
ian.abraham@yale.edu

**Fabio Ramos**
Nvidia, USA
The University of Sydney, Australia
fabio.ramos@sydney.edu.au

## Abstract

Effective robotic exploration in continuous domains requires planning trajectories that maximize coverage over a predefined region. A recent development, Stein Variational Ergodic Search (SVES), proposed parallel ergodic exploration (a key approach within the field of robotic exploration), via Stein variational inference that computes a set of candidate trajectories approximating the posterior distribution over the solution space trajectories. While this approach leverages GPU parallelism well, the trajectories in the set might not be distinct enough, leading to a suboptimal set. In this paper, we propose two key methods to diversify the solution set of this approach. First, we leverage the signature kernel within the SVES framework, introducing a pathwise, sequence-sensitive interaction that preserves the Markovian structure of the trajectories and naturally spreads paths across distinct regions of the search space. Second, we propose a derivative-free evolution-strategy interpretation of SVES that exploits batched, GPU-friendly fitness evaluations and can be paired with approximate gradients whenever analytic gradients of the kernel are unavailable or computationally intractable. The resulting method both retains SVES's advantages while diversifying the solution set and extending its reach to black-box objectives. Across planar forest search, 3D quadrotor coverage, and model-predictive control benchmarks, our approach consistently reduces ergodic cost and produces markedly richer trajectory sets than SVES without significant extra tuning effort.

## 1 Introduction

Robotic exploration–the autonomous process by which robots survey unknown or partially mapped environments to collect information, build maps, and accomplish tasks such as search-and-rescue, environmental monitoring, or infrastructure inspection. The success of robotic exploration hinders on the reliability of the robot to adapt its exploratory opteration in dynamic or unstructured settings where pre-programmed paths may fail. A key approach within this field is ergodic search [1, 2], in which trajectories are generated so that the robot's time-averaged visitation frequency matches a desired spatial distribution of information. By ensuring that time spent in each region is proportional to its importance, ergodic search provides systematic, efficient coverage of the domain and avoids both redundantly revisiting well-known areas and neglecting critical regions. In such applications where a robot must systematically explore an area, having multiple, robust, high-quality and diverse set of trajectories is critical to adapt to dynamic environments and quickly switch between search

39th Conference on Neural Information Processing Systems (NeurIPS 2025).

strategies. Moreover, a set of diverse solutions aid in avoiding "bad" local minima and lead to a better optimization outcome.

Recent progress in curiosity- and information-based exploration has enabled robots to survey large, unstructured environments [3–7]. However, most existing methods rely exclusively on an information-maximizing strategy, which often leads to myopic behavior: agents greedily seek immediate gains in information without regard for long-term, strategically advantageous states.

Ergodic exploration techniques address this limitation by casting exploration as a coverage problem, optimizing time-averaged trajectory statistics to ensure sustained visitation of high-information regions [8, 9]. Concretely, these methods optimize the spatial distribution of trajectory dwell time against an expected information measure, allowing robots to plan exploratory paths that effectively handle multi-modal search landscapes [1]. Despite their demonstrated effectiveness, current ergodic algorithms still optimize only a single search policy at a time, constraining a robot's ability to adapt when tasks or environments change.

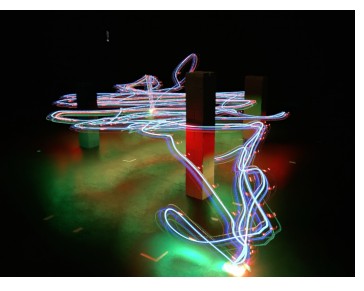 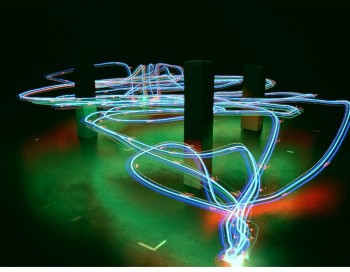

(a) RBF kernel        (b) Signature kernel

Figure 1: illustrates the qualitative impact of our signature-kernel enhancement: where trajectories generated using the RBF kernel (panel a) not only cluster around the domain center but also produce rough, lower-quality paths, the signature kernel (panel b) yields smooth, high-quality trajectories that spread more uniformly across the entire domain, achieving significantly richer ergodic coverage. These trajectories were flown by Crazyflie drones in a real-world setting as part of the experiment described in Section 7.2.

Prior work has observed that the non-convex nature of the ergodic objective can yield multiple locally optimal trajectories under different initializations [9]. Yet explicitly reasoning over—and sampling from—a distribution of such trajectories is computationally prohibitive. Moreover, there is no assurance that distinct initial conditions will avoid collapsing onto the same ergodic solution, a critical shortcoming in online exploration scenarios where mode collapse can incur catastrophic performance failures.

Stein variational inference methods show promise in providing the necessary tools to approximate distributions of trajectories in a computationally tractable manner [10]. Motivated by these strengths, [2] proposes Stein variational ergodic search (SVES), a formulation of ergodic exploration as a Stein variational inference problem: by applying Stein variational gradient descent to the space of robot trajectories. However, SVES neither fully exploits the inherent Markovian structure of trajectories nor avoids a strong dependence on gradient information. Moreover, although SVES alludes to promoting diversity among trajectories, it does not concretely define this notion or provide what "diversity" entails.

**Contributions:** We address the challenges mentioned above by "diversifying" Stein Variational Ergodic Search (SVES). Our approach is built on two complementary pillars. First, we leverage the signature kernel similar to [11], enabling us to encode the intrinsic sequential (or Markovian) and geometric properties of trajectories. The signature kernel naturally discriminates between subtly different paths, thus promoting diversity. Second, we incorporate Stein Variational Covariance Matrix Adaptation Evolution Strategy (SV-CMA-ES) [12] and Simultaneous Perturbation Stochastic Approximation (SPSA) [13] into SVES, drawing on recent progress in Stein Variational Evolution Strategies, this approach not only eliminates the need for gradients—often costly to compute—by harnessing GPU parallelism, making it viable for real-time applications, but also enables the use of complex non-differentiable or black-box loss functions and constraints. Later we show empirically that this approach leads to better optimization results and a more diverse set of solutions.

**Paper organization:** The remainder of the paper is organized as follows. In Section 2 we formalize the ergodic search problem, Section 3 reviews Stein variational ergodic search, Section 4 introduces

the signature transform and signature kernel, Sections 5 and 6 present our diversification strategies using the signature kernel and SV-CMA-ES respectively, Section 7 reports empirical results, Section 8 concludes and discusses the limitations and best practices.

## 2 Problem Statement

We consider a robot with state space $\mathcal{X} \subseteq \mathbb{R}^n$ and control space $\mathcal{U} \subseteq \mathbb{R}^m$, whose continuous-time dynamics are

$$\dot{x}(t) = f\big(x(t),\, u(t)\big), \quad x(0) = x_0 \in \mathcal{X},\ u : [0, T] \to \mathcal{U}.$$

A robot's space trajectory $x(t) : [0, T] \to \mathcal{X}$ is defined as the solution of the corresponding initial-value problem (integrated over time):

$$x(t) = x(0) + \int_0^t f\big(x(\xi),\, u(\xi)\big)\, d\xi. \tag{1}$$

We denote the set of all such finite-horizon trajectories by $\tau = \bigcup_{T>0} \mathcal{C}^0\big([0, T],\, \mathcal{X}\big)$, as the set of all continuous trajectories defined on arbitrary finite time intervals. [1] The robot explores a bounded domain $\mathcal{S} = [0, B_0] \times \cdots \times [0, B_{v-1}] \subset \mathbb{R}^v, \quad v \leq n$, via a projection $g : \mathcal{X} \to \mathcal{S}$.

We assign each region of $\mathcal{S}$ an *importance* according to a target probability measure $\pi$. An *ergodic cost* $\mathcal{E}_\pi(x)$ intuitively measures how well a trajectory $x \in \tau$ covers $\mathcal{S}$ in proportion to $\pi$: it penalizes regions that are under- or over-visited relative to their importance. [2] Driving $\mathcal{E}_\pi(x) \to 0$ therefore ensures the robot spends time in each region proportional to its assigned importance (see Definition 5 in Appendix D a formal introduction on ergodic cost). Our objective is to compute feasible trajectories that minimize this ergodic cost while satisfying the constraints of the system. Over a fixed horizon $T$, we solve

$$\begin{aligned} \underset{\substack{x(t) \in \tau(\mathcal{X}) \\ u(t) \in \mathcal{U},\, \forall t \in [0, T]}}{\text{minimize}} \quad & \mathcal{E}_\pi(x(t)) \tag{2} \\ \text{subject to} \quad & h_1(x) = 0, \quad h_2(x) \leq 0, \quad \forall t \in [0, T] \end{aligned}$$

where initial conditions $x(0)$ and trajectory constraints can be accounted for in the equality and inequality constraints $h_1, h_2$, e.g., $\dot{x} = f(x, u)$ and $u \in \mathcal{U}$.

In what follows, we develop Stein-variational and evolutionary-strategy methods to efficiently generate diverse high-quality ergodic trajectories in parallel.

## 3 Stein variational ergodic search

SVES reformulates ergodic search (a problem where a robot must explore the domain $\mathcal{S}$ according to a target distribution, see Appendix D) as an inference problem. A binary indicator $O : \tau(\mathcal{X}) \to \{0, 1\}$, flags trajectories that satisfy the optimality criterion. The likelihood of $O = 1$ given a trajectory $x$ is defined by $p\big(O = 1 \mid x\big) = \exp\big(-\mu\, \mathcal{E}_\pi(x)\big)$, where $\mathcal{E}_\pi(x)$ is the ergodic cost and $\mu > 0$ scales its influence. By combining this likelihood with a prior $p(x)$ over the space of trajectories via Bayes' rule, one can obtain the posterior $p(x \mid O)$. We use Stein Variational Gradient Descent (SVGD) [10] (see Appendix B), an iterative algorithm to approximate this posterior $p(x \mid O) \propto p(x) \exp\big(-\mu\, \mathcal{E}_\pi(x)\big)$ with a set of particles so that their empirical distribution approximates the target posterior. At each iteration, particles are nudged toward to regions of high posterior density and while being repelled from one another to promote coverage, thus forming a sample-based approximation of the true posterior

**Integrating dynamics and constraints:** The dynamics and constraints $h_1(x) = 0$, $h_2(x) \leq 0$, etc., can be folded into a composite cost

$$\mathcal{J}_\pi(x) = \mathcal{E}_\pi(x) + \rho(x) + c_1\, h_1(x)^2 + c_2 \max\{0, h_2(x)\}, \tag{3}$$

---

[1] Throughout this paper we use the notation $C^x(I, V) := \{f : I \to V \mid f^{(j)} \text{ exists and is continuous on } I \text{ for all } 1 \leq j \leq x\}$, and $C^0(I, V) := \{f \text{ is continuous}\}$.

[2] Since time is a limited resource, it is crucial to penalize excessive revisitations in order to maximize the useful information gathered within a finite time budget.

with $\rho$ additional penalties and $c_i > 0$. The likelihood is then given by $p(x) = \exp\big(-\mu\,\mathcal{J}_\pi(x)\big)$. Throughout this paper, we set $\mu = 1$.

## 3.1 SVES over state trajectories

We discretize a continuous trajectory $x(t)$ into $\mathbf{x} = [x_0, \ldots, x_{T-1}]$, and chose a Gaussian prior $p(\mathbf{x}) = \mathcal{N}(\hat{x}, \sigma^2 I)$, where $\hat{x}$ interpolates boundary states. SVGD maintains $N$ "particles", in our case $N$ trajectories $\{\mathbf{x}_r^i\}_{i=1}^N$ and updates them via

$$\phi_r^*(\cdot) = \frac{1}{N}\sum_{i=1}^N[k(\mathbf{x}_r^i, \cdot)(\nabla_{\mathbf{x}}\log p(\mathbf{x}_r^i) - \mu\nabla_{\mathbf{x}}\mathcal{J}_\pi(\mathbf{x}_r^i)) + \nabla_{\mathbf{x}}k(\mathbf{x}_r^i, \cdot)], \tag{4}$$

where $r$ is an arbitrary SVGD iteration. The first SVGD term pulls particles toward low-cost ergodic trajectories (or modes with high density) and is often referred to as the attractive force; the second, the gradient of the kernel disperses them to preserve coverage and is referred to as the repulsive force. Therefore, the choice of an effective kernel is essential for good mode coverage. A key insight to note is that SVGD allows the $N$ particles to be updated in parallel.

## 3.2 SVES over control sequences

Similar to before, one can optimize over discrete control sequences $\mathbf{u} = [u_0, \ldots, u_{T-1}]$, with the discretized dynamics $x_{t+1} = F(x_t, u_t)$, $x_0$ given, and a prior $p(\mathbf{u})$ using SVGD and each control-particle $\mathbf{u}_r^i$ is updated by

$$\phi_r^*(\cdot) = \frac{1}{N}\sum_{i=1}^N[k(\mathbf{u}_r^i, \cdot)(\nabla_{\mathbf{u}}\log p(\mathbf{u}_r^i) - \mu\nabla_{\mathbf{u}}\mathcal{J}_\pi(\mathbf{x}_r^i|\mathbf{u}_r^i, x_0)) + \nabla_{\mathbf{u}}k(\mathbf{u}_r^i, \cdot)], \tag{5}$$

In an MPC-style loop, usually only the first control action is applied, the state is estimated, and the SVGD optimization is repeated for the shifted horizon.

See Appendix E for a more detailed introduction to SVES.

## 4 Signature Transform and Signature Kernel

The signature (transform) and its corresponding kernel [14] have proven to be highly effective in diverse applications—ranging from robotics to deep learning—where data is naturally represented as a continuous path [14–19]. Informally, the signature transform of a trajectory can be viewed as analogous to the Fourier series, where the trajectory is represented as a countably infinite sequence. We now introduce the signature (transform) of a path/trajectory and its corresponding kernel:

**Definition 1. *(Informal) Signature (transform) of a path [20].* *Consider a continuous path* $x : I \to \mathbb{R}^d$ *on an interval* $I \subset \mathbb{R}$. *For any subinterval* $[s, t] \subset I$, *the* signature *of* $x$ *over* $[s, t]$ *is defined as the sequence***

$$\varphi(x)_{s,t} = \Big(1, \int_{s<u_1<t} dx(u_1), \ldots, \int_{s<u_1<\cdots<u_k<t} dx(u_1) \otimes \cdots \otimes dx(u_k), \ldots\Big), \tag{6}$$

*where* $\otimes$ *denotes the classical tensor product. This collection of iterated integrals captures the essential features of the path and is invariant under reparameterization (i.e.,*$\varphi(x) = \varphi(x \circ \theta)$, *for some reparameterization* $\theta$).

Essentially one can view the path signature is a canonical linear embedding of any multivariate path into an (infinite) series of iterated integrals. It is injective on all non-tree-like paths, meaning no two such paths share the same signature (in practice one ensures non-tree-likeness by appending a monotonic "time" coordinate). It also satisfies *time-reversal duality*: $\varphi(x)_{a,b} \otimes \varphi(\overleftarrow{x})_{a,b} = 1$, where 1 is the unit element and $\overleftarrow{x}(t) = x(a + b - t)$.

One can approximate signature kernel reasonably well with the *truncated signature transform* of level $L \in N$ and [21, 22] provide efficient means to compute the truncated signature using Dynamic Programming and approximate it using Random Fourier Features.

**Definition 2.** *(Informal) Signature Kernel* *Let $I = [u, u']$ and $J = [v, v']$ be two intervals, and consider paths $x \in \mathcal{C}^1(I, \mathbb{R}^d)$ and $y \in \mathcal{C}^1(J, \mathbb{R}^d)$. The signature kernel $k^{sig}(x, y) : I \times J \to \mathbb{R}$ is defined by $k^{sig}_{s,t}(x, y) = \langle \varphi(x)_s, \varphi(y)_t \rangle$, where $\varphi(x)_s$ and $\varphi(y)_t$ denote the signatures of $x$ and $y$ over the intervals $[u, s]$ and $[v, t]$, respectively.*

Utilizing the *kernel trick* introduced in [20], one can efficiently compute the full (untruncated) signature kernel. The kernel trick uses the following result:

**Theorem 1.** *[20] (Informal) Let $I = [u, u']$ and $J = [v, v']$ be 2 intervals, and let $x \in \mathcal{C}^1(I, \mathbb{R}^d)$ and $y \in \mathcal{C}^1(J, \mathbb{R}^d)$. Then the signature kernel $k_{x,y}$ satisfies the following partial differential equation:*

$$\frac{\partial^2 k^{sig}_{s,t}}{\partial s\, \partial t}(x, y) = \langle \dot{x}(s),\, \dot{y}(t) \rangle_V\, k_{s,t}(x, y),$$

*subject to the boundary conditions $k^{sig}_{u,t}(x, y) = 1 \quad \forall t \in J, \qquad k^{sig}_{s,v}(x, y) = 1 \quad \forall s \in I.$*

The resulting PDE formulation of the signature kernel enables its efficient computation using any standard numerical scheme, such as finite difference, finite element, or other suitable methods for hyperbolic equations. See Appendix G.5 for a formal introduction to the signature kernel and the reasons why it makes a good feature map for trajectories.

## 5 Promoting diversity in SVES using signature kernels

The kernels used in the previous work on SVES [2] do not take full advantage of the Markovian property of the trajectories. Hence, we propose to directly apply $k^{sig}$ in equations 4 and 5. Computing the full $(N \times N)$ Gram matrix for $N$ trajectories of length $T$ in $\mathbb{R}^d$ via dynamic programming incurs a time complexity of $O(N^2 T^2 d^c)$, where $c$ depends on the level of truncation of the signature transform. By exploiting the kernel trick as stated in Theorem 1, the cost of evaluating the kernel between any two paths can computed in as low as $O(T^2 d)$. Although forming the complete Gram matrix still requires $N^2$ such evaluations and hence retains an $O(N^2 T^2 d)$ overall cost, the underlying PDE formulation admits efficient parallelization. In particular, when implemented with an explicit finite–difference scheme on a suitably provisioned GPU, the per-pair computation can be driven down to linear dependence on the trajectory length (in practice), yielding an effective wallclock complexity of $O(T\, d)$ under the assumption that the GPU can accommodate the required number of threads [20].

## 6 Promoting Diversity in SVES using SV-CMA-ES

CMA-ES updates have been demonstrated to yield more efficient search steps than standard gradient descent across a variety of benchmark problems [23–25]. Moreover, [26] empirically established that SV-CMA-ES, a variant of SVGD that uses CMA updates, can surpass SVGD in update efficiency, leading to faster and more accurate convergence to the ground truth under certain conditions. In the spirit of these findings, we integrate SV-CMA-ES into the SVES framework in lieu of SVGD. A background in SV-CMA-ES can be found in Appendix C. SVGD's update requires computing the gradient of the log density $\nabla \log p(x)$ (a process that typically involves multiple sequential back-and-forth evaluations of the log-density, in practice) which can be very expensive [10]. However, SV-CMA-ES updates involve the evaluation of a *fitness function* via many independent samples, which turn is used to compute $\Delta_j \approx \sigma_j \sum_{k=1}^{m} w_{j,k}\, y_{j,k}$. a gradient-free surrogate of $\nabla \log p(x)$ (see C Equation 11). Consequently, on massively parallel hardware (e.g., GPUs), SV-CMA-ES may reduce wall-clock time compared to gradient-based SVGD, in cases where computing the gradient of the log density is very expensive or not available. In the case of Ergodic search, the fitness function $f(\cdot)$ in the update (Equation 12) is instantiated as the ergodic cost $\mathcal{J}_\pi(\cdot)$.

Moreover, SV-CMA-ES's gradient-free exploration–exploitation scheme further enhances trajectory diversity: it simultaneously samples across multiple subpopulations (exploration) and then refines its search by exploiting a select subset of high-fitness (elite) trajectories.

**Evolution Strategies for Kernel Gradient Estimation:** When the analytical gradient of the kernel is not readily available or is computationally intractable, Simultaneous Perturbation Stochastic Approximation (SPSA) may be employed. SPSA is a gradient-free technique for estimating gradients

of black-box functions via simultaneous random perturbations [13]. SPSA, similar to SV-CMA-ES performs multiple, independent kernel evaluations in parallel, thus can also exploit GPU parallelism to deliver lower overall wall-clock execution times. We found that the noise from SPSA, in some cases aids in breaking out of a "bad" local minima, but in most cases provide no significant performance benefits. By coupling SV-CMA-ES with SPSA, we obtain a fully gradient-free algorithm for generating ergodic trajectories, which is the method adopted in our experimental evaluation (and hence we will refer to SV-CMA-ES coupled with SPSA as just SV-CMA-ES from this point on).

# 7   Experimental results

In this section, we present results to empirically demonstrate the effectiveness and applicability of the methods discussed above in a variety of simulated and real experiments (and their exact setup can be found in Appendix K). Specifically, we are interested in addressing the following:

1. How diverse are the trajectories produced using the signature kernel?
2. How ergodic are the trajectories produced with the signature kernel?
3. What are the convergence rates of SV-CMA-ES and SVGD?
4. How ergodic and diverse are the trajectories produced using SV-CMA-ES compared to SVGD?

To address 1) and 2) we will benchmark the signature kernel with other well-known kernels, specifically, the RBF kernel, kernelized DTW, Global Alignement kernel and the Markov RBF kernel. A brief description of these kernels can be found in Appendix G. At each iteration of the SVGD or SV-CMA-ES a new bandwidth for the kernels was chosen using the median heuristic. We employed the median heuristic for bandwidth selection because it is straightforward, computationally light, and requires no manual tuning, making our method more accessible to users (refer to [27] for details on the guarantees provided by the median heuristic).

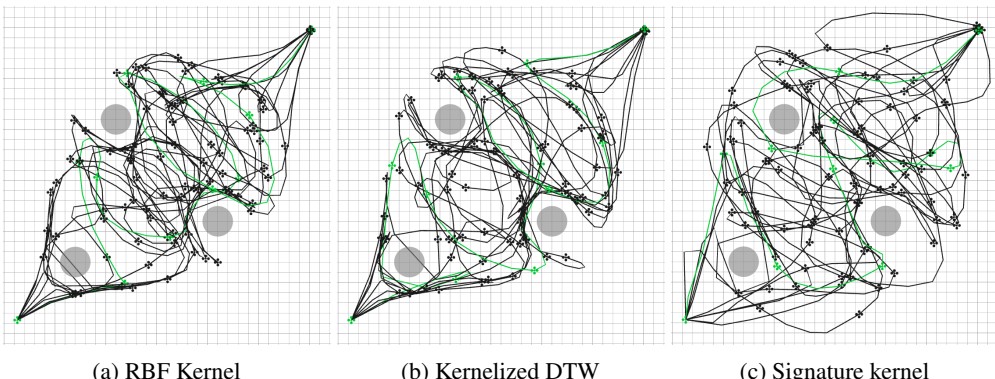

|              (a) RBF Kernel              |           (b) Kernelized DTW           |           (c) Signature kernel           |

Figure 2: Top down projection of trajectories generated in experiment comparing kernels in section 7.2.1. Despite using a Gaussian prior that just interpolates start and goal, only the signature kernel's strong repulsive term steers trajectories around obstacles.

The diversity of $N$ trajectories is measured by $\det(K_f)$, where $K_f \in \mathbb{R}^{N \times N}$ is the Gram matrix with entries $(K_f)_{ij} = k_{\text{Fréchet}}(\mathbf{x}_i, \mathbf{x}_j)$, where $k_{\text{Fréchet}}$ is kernel build using the Fréchet distance. See Appendix H for details.

## 7.1   Multiscale constrained exploration

### 7.1.1   Signature kernel performance

To evaluate the performance of the signature kernel within the SVES framework (using SVGD), we instantiate ten distinct priors $p_k(\cdot)$ for $k = 1, \ldots, 10$ as detailed in Section 3.1 & E.2, each with variance $\sigma^2 = 0.01$. For each prior, we generate $N = 6$ trajectory samples and compare the kernels according to the following criteria in SVES:

1) Average trajectory cost, $\frac{1}{N}\sum_{i=1}^{N}\mathcal{J}_\pi(\mathbf{x}^i)$.
2) Best trajectory cost, $\min_{1\leq i\leq N}\mathcal{J}_\pi(\mathbf{x}^i)$.
3) Mean ergodic cost, $\frac{1}{N}\sum_{i=1}^{N}\mathcal{E}_\pi(\mathbf{x}^i)$.
4) Trajectory diversity. $-\log\big(1-\det(K_f)\big)$

All Stein variational gradient updates employ a step size of $\epsilon = 0.01$, and iterations are terminated when $\big\|\mathbf{x}_{(i)} - \mathbf{x}_{(i-1)}\big\| < 1.25 \times 10^{-3}$ or maximum of 3000 iterations is reached. The experimental results are summarized in Figure 3 and an elaborate statistical analysis can be found in Appendix J (Table 2).

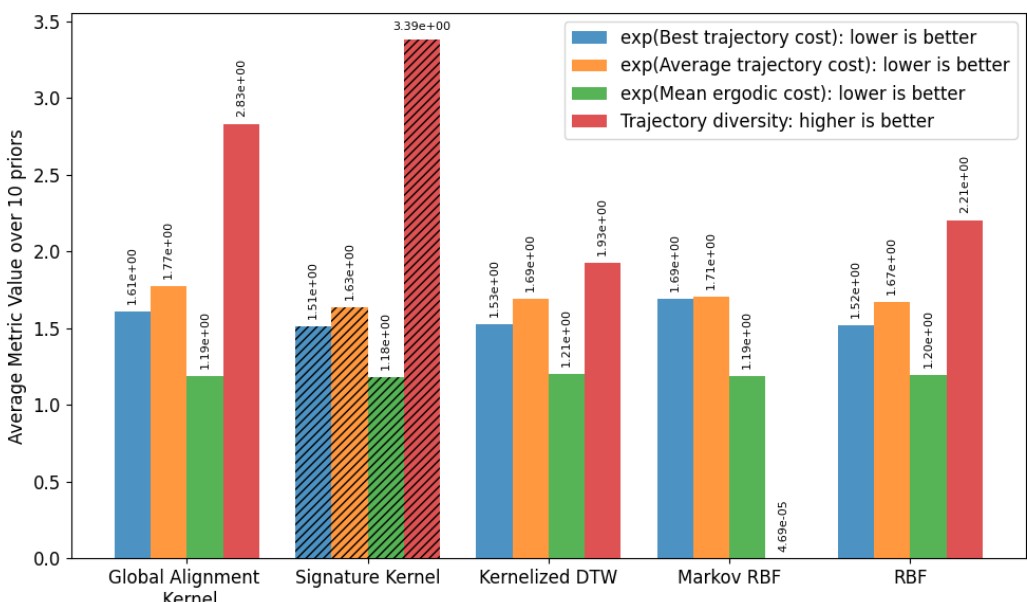

Figure 3: Aggregated performance comparison across five kernel categories. For each kernel, the four side-by-side bars represent the average of 10 experimental runs for the metrics discussed in 7.1.1.

### 7.1.2 SV-CMA-ES vs SVGD

To assess the comparative performance of SV-CMA-ES and standard SVGD on the ergodic trajectory optimization, we initialize both methods with $N = 5$ trajectories drawn from the same Gaussian prior (variance $\sigma^2 = 0.01$). All other hyperparameters are held constant. Here we use the RBF kernel. At each optimization iteration, we record:

1) Average trajectory cost, $\frac{1}{N}\sum_{i=1}^{N}\mathcal{J}_\pi(\mathbf{x}^i)$.
2) Best trajectory cost $\min_{1\leq i\leq N}\mathcal{J}_\pi(\mathbf{x}^i)$.
3) Trajectory diversity, $-\log\big(1-\det(K_f)\big)$

Figure 4 illustrates the per-iteration evolution of these metrics. SV-CMA-ES maintains similar costs compared to SVGD, while promoting trajectory diversity.

## 7.2 3D Coverage using the Crazyflie drone

### 7.2.1 Signature kernel performance

We follow the exact method used in section 7.1.1, but in example here set $N = 10$, $\epsilon = 0.02$, and iterations are terminated when $\big\|\mathbf{x}_{(i)} - \mathbf{x}_{(i-1)}\big\| < 1.25 \times 10^{-3}$ or a maximum of 1500 iterations is reached.

The experimental results are summarized in Figure 5 and an elaborate statistical analysis can be found in Appendix J (Table 3).

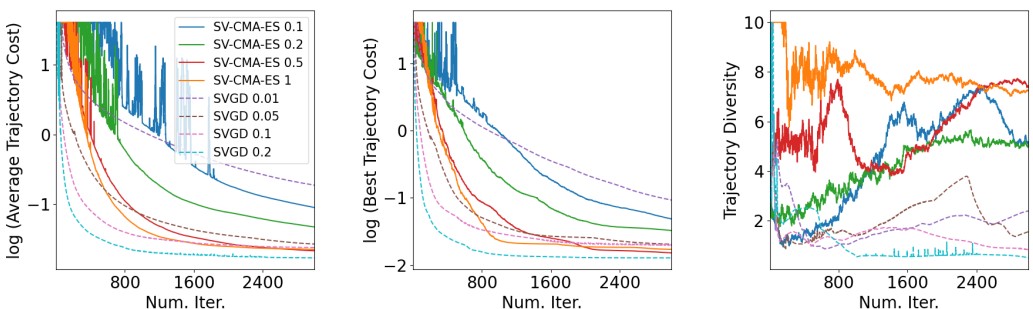

Figure 4: Convergence results for different step sizes (defined as $\epsilon$ for SVES and $\alpha_x$ for SV-CMA-ES in Appendices B & C respectively) for the multiscale constrained exploration experiment.

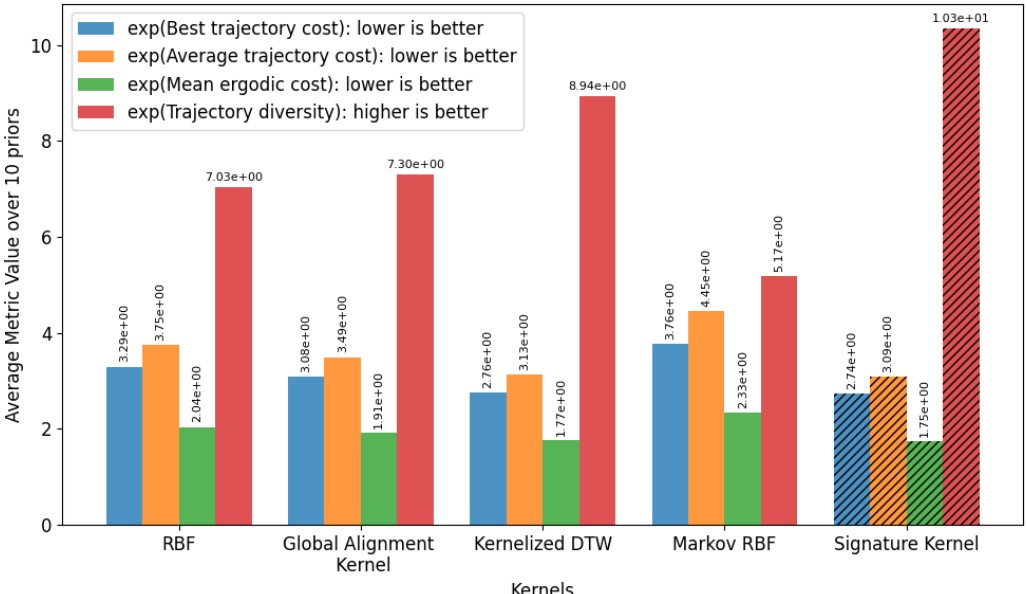

Figure 5: Aggregated performance comparison across five kernel categories. For each kernel, the four side-by-side bars represent the average of 10 experimental runs for the metrics discussed in 7.2.1.

### 7.2.2 SV-CMA-ES vs SVGD

We once again follow the method in section 7.1.2 with $N = 5$ trajectories. Figure 6 illustrates the per-iteration evolution of these metrics.

## 7.3 Stein variational model-predictive control

### 7.3.1 Signature Kernel Performance

We instantiate ten distinct priors $p_k(\cdot)$ for $k = 1, \ldots, 10$ similar to 7.1.1. For each prior, we record the following at each model predictive control (MPC) iteration $j = 1, \ldots, 200$.

1) Average trajectory cost,
$$\bar{\mathcal{J}}^j := \frac{1}{N} \sum_{i=1}^{N} \mathcal{J}_\pi^j(\mathbf{u}^i).$$

2) Best trajectory cost,
$$\mathcal{J}^{j,*} := \min_{1 \le i \le N} \mathcal{J}_\pi^j(\mathbf{u}^i).$$

3) Ergodic cost of the best trajectory,
$$\mathcal{E}^{j,*} := \mathcal{E}_\pi^j(\mathbf{u}_i) :$$
$$i = \arg\min_{k \in \{1,\ldots,N\}} \mathcal{J}_\pi^j(\mathbf{u}_k).$$

4) Trajectory diversity,
$$D^j := -\log\left(1 - K_f^j\right)$$

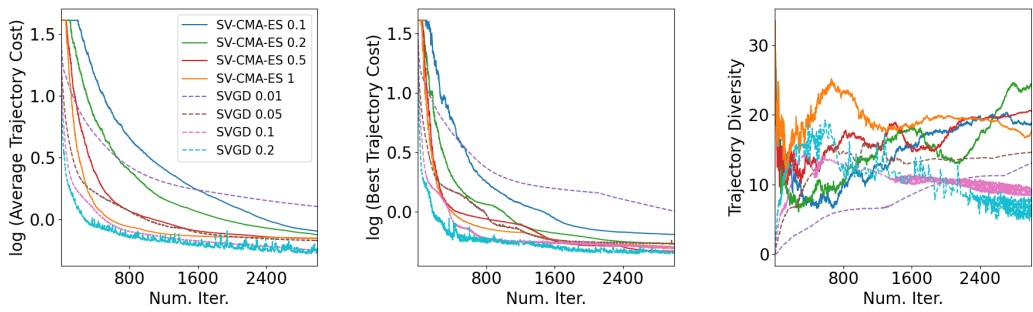

Figure 6: Convergence results for different step sizes (defined as $\epsilon$ for SVES and $\alpha_x$ for SV-CMA-ES in Appendices B & C respectively) for the 3D coverage using the Crazyflie drone experiment.

Table 1: Aggregated metrics over all time steps for a single prior

|  | RBF | Signature kernel | Markov RBF | Kernelized DTW |
|---|---|---|---|---|
| Total average trajectory cost: $\sum_{j=1}^{200} \bar{\mathcal{J}}^j$ | 563.14 | 581.27 | 764.08 | 554.92 |
| Total best trajectory cost: $\sum_{j=1}^{200} \mathcal{J}^{j,*}$ | 377.83 | 351.83 | 468.82 | 375.13 |
| Total ergodic cost of best trajectory: $\sum_{j=1}^{200} \mathcal{E}^{j,*}$ | 262.09 | 239.73 | 319.88 | 262.09 |
| Total trajectory diversity: $\sum_{j=1}^{200} D^j$ | 48.51 | 548.22 | 3395.45 | 57.46 |

The experimental results for one such prior are summarized Table 1. The experimental results for all 10 priors are summarized in Appendix J Table 4.

Markov RBF failed to converge to a meaningful solution in most of the MPC iterations—resulting in essentially random trajectories and thus artificially high diversity values.

## 7.4 Ablations: kernel bandwidth sensitivity

We provide a small ablation study on the effect of the choice of bandwidth on kernels in the 3D Coverage using the Crazyflie drone experiment from Section 7.2. Figure 7 compares 3 different computationally light heuristics [28] (mean heuristic, Sliverman's rule of thumb and Scott's rule) for selecting the bandwidth and Figure 8 shows the effect of manually tuning the kernel.

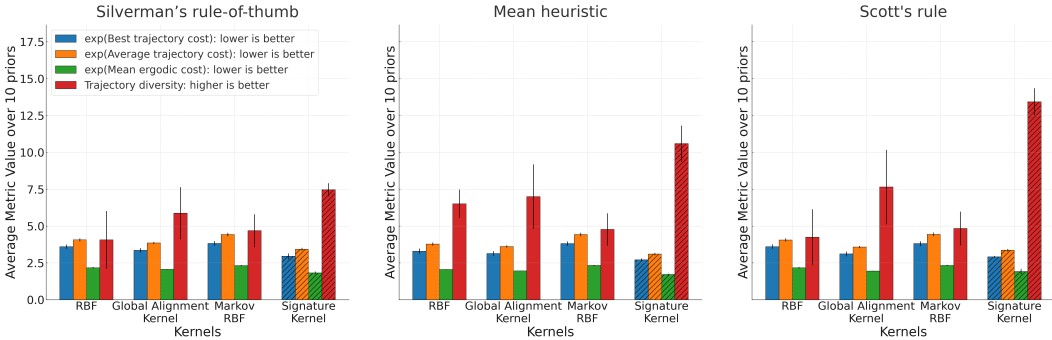

Figure 7: The 3D coverage using the Crazyflie drone experiment with different kernel bandwidth selection heuristics. The plots report the mean over 10 random priors, with error bars indicating $\pm 1$ standard deviation.

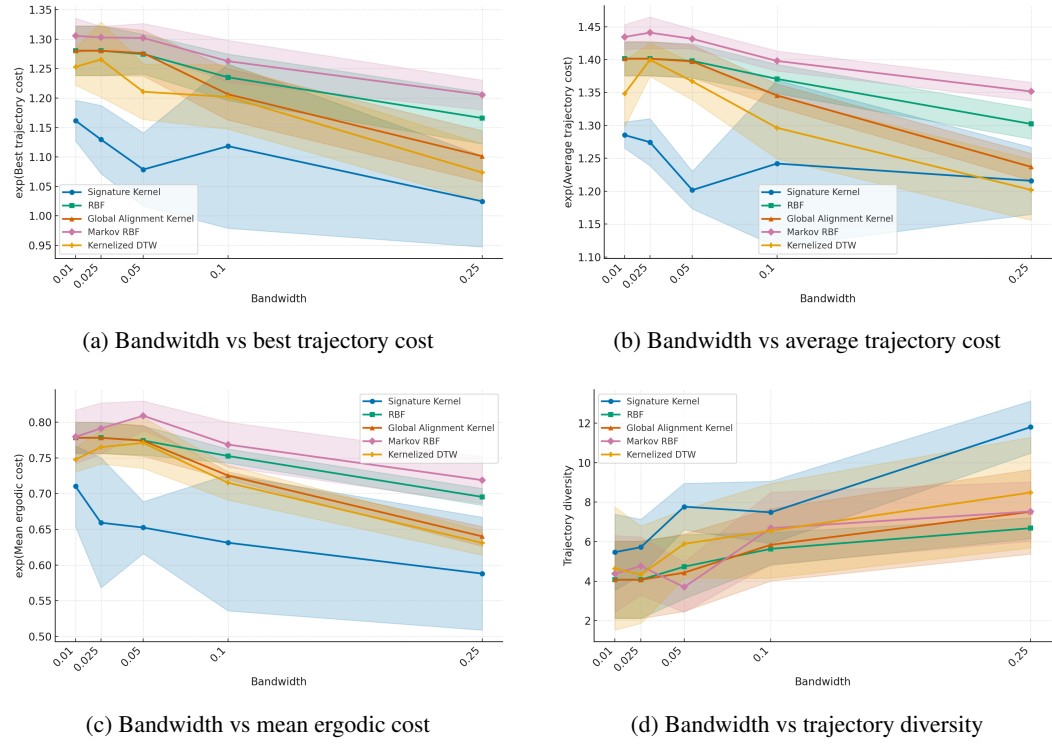

(a) Bandwitdh vs best trajectory cost

(b) Bandwidth vs average trajectory cost

(c) Bandwidth vs mean ergodic cost

(d) Bandwidth vs trajectory diversity

Figure 8: Impact of bandwidth selection on kernel performance in 3D coverage using the Crazyflie drone experiment. Each sub-figure presents the mean over 5 random priors, with error bars indicating $\pm 1$ standard deviation.

## 8    Conclusion, Limitations and best practices

In summary, integrating the signature kernel into our ergodic search framework consistently lowers ergodic cost and increases trajectory diversity, compared to standard choices by promoting more effective mode coverage. Moreover, swapping in SV-CMA-ES further enhances exploration diversity, while completely being a gradient-free approach, underscoring the practical benefits of our approach. See Appendix J for the visual representation of the trajectories.

However our approach has several limitations. First, the runtime of the methods discussed in this paper depends heavily on the available CPU/GPU parallelism and the dimensionality of the problem: as the number of particles and dimensions increases, both communication and computation overhead grow, which can degrade performance on hardware with limited cores/threads or when scaling to very high dimensions. Second, the SV-CMA-ES instantiation usually requires performing a singular value decomposition (SVD) or Cholesky factorization at every iteration. Consequently, the scalability (and hence the real time applicability) of this method depends heavily on the problem size, use case, cost of computing the gradients, available hardware etc., and practitioners should be aware of these trade-offs. Future work could explore incorporating recent advances (e.g., diagonal covariance matrices [29]) in vanilla CMA to improve efficiency. Third, while the signature kernel provides a very strong repulsive force that promotes diverse exploration, it can also inadvertently push trajectories into obstacles, outside bounded domains, etc. We noticed this specially in the example in section 7.3, which explains the higher total average trajectory cost. To mitigate these effects, practitioners should consider using more conservative cost designs when employing the signature kernel, opting for alternative kernels (e.g., RBF) that exert milder repulsion or running SVGD for more iterations (see Appendix F for the exact convergence guarantees; we also recommend the reader to refer to [2, 30]). Fourth, SV-CMA-ES does not provide rich theoretical guarantees provided by SVGD and hence SVES. Finally, we did not explore the potential of the Fréchet Distance kernel for producing diverse trajectories due to its computational cost.

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

# Appendices

## Appendix Contents

## A  Additional related work

**Trajectory optimization**   Trajectory optimization encompasses a wide range of techniques that initiate with a preliminary, suboptimal path and iteratively refine it to converge on a (potentially local) optimum by minimizing an associated cost function. Foundational contributions in this domain include Covariant Hamiltonian Optimization for Motion Planning (CHOMP) [31] and its related approaches [32–34], which exploit the covariance of trajectories alongside Hamiltonian Monte Carlo to perform annealed functional gradient descent. A significant drawback of these methods is their reliance on fully differentiable cost functions. Stochastic Trajectory Optimization for Motion Planning (STOMP) [35] overcomes this by estimating gradients from stochastic samples of noisy trajectories, thereby accommodating cost functions that lack differentiability.

Another influential research direction involves quality diversity algorithms, notably the Covariance Matrix Adaptation Evolution Strategy (CMA-ES) [36–38], which adopts a derivative-free strategy using multivariate normal distributions to generate and update candidate solutions. Although CMA-ES is robust and exhibits ergodic behavior capable of managing multimodal problems, it typically requires multiple initializations and a higher number of evaluations compared to gradient-based methods [39]. Other techniques, such as TrajOpt [40], solve sequential quadratic programs with continuous-time collision checking; while these methods are efficient, they often yield only locally optimal solutions and necessitate fine parametrization of waypoints. To mitigate this, [34] limit the optimization to a Reproducing Kernel Hilbert Space (RKHS) with a squared-exponential kernel, although this may overlook the cost contributions between sparsely distributed waypoints. Methods proposed in GPMP [41–43] model trajectories as Gaussian Processes (GP) to obtain a maximum a posteriori (MAP) solution.

More closely aligned with our approach, variational inference methods introduced in [44] and [45] infer the posterior distribution over trajectories using Stein variational gradient descent (SVGD) or

natural gradient updates. In contrast, the RKHS induced in SVMP fails to capture the sequential dynamics of trajectories by treating them as static, high-dimensional vectors—resulting in reduced repulsive forces and less coordination across functional dimensions. Kernel Signature Variational Gradient Descent (SigSVGD) [11], overcomes these shortcomings by integrating path signatures to explicitly encode the sequential structure of trajectories. This approach not only enhances diversity but also increases the probability of discovering superior local optima. Furthermore, leveraging insights from both motion planning and control-as-inference allows SigSVGD to extend naturally to Model Predictive Control (MPC) frameworks (e.g., Stein Variational MPC (SVMPC) [46]), where preserving the sequential structure of control trajectories is crucial.

**Robotic exploration**  Robotic exploration is primarily concerned with steering agents into unknown territories, while the coverage problem entails designing trajectories and strategically placing sensors to ensure that a confined area is thoroughly examined. Early solutions often relied on discrete grid-based strategies—employing methods such as advanced boustrophedon search patterns (the antecedent to the lawnmower pattern [47]) and variations of the traveling salesperson problem [48, 49]—to list all viable exploration routes. Although grid-based methods offer guarantees of completeness, scaling these techniques to continuous domains is inherently difficult because the number of potential trajectories becomes unbounded. Recent strategies in reinforcement learning and information-driven exploration have sought to surmount these obstacles by using "curiosity" measures [3, 4] derived from information theory [50]. However, many of these approaches tend to be short-sighted, concentrating on immediate rewards and often settling on a single-mode solution, which restricts the system's overall flexibility.

Subsequent developments have brought forth ergodicity-based methods for exploration. By reducing the disparity between the expected spatial information distribution and the time-averaged visitation of a path [8], ergodic exploration techniques yield sophisticated coverage patterns across continuous domains [1, 51]. In fact, the spectral, multi-scale structure of the ergodic metric [52] suggests the existence of numerous exploitable local solutions—an element that is essential for resilient, real-time exploration [53]. Despite these advances, a persistent challenge remains in devising methods that compute sets of "good," locally optimal ergodic trajectories which can facilitate efficient strategy switching in dynamic scenarios.

**Control as inference**  The control as inference framework has recently emerged as a powerful perspective by recasting trajectory optimization issues within a probabilistic framework. The inherent non-convexity of many robotic problems—due to complex dynamics and contact interactions—creates significant hurdles for traditional gradient-based optimization techniques. Approaches such as predictive sampling [54] and model-predictive path integral control (MPPI) [55, 56] have been developed to mitigate these difficulties. However, these methods often produce only a single locally optimal solution, thereby overlooking the potential for multiple, equally effective strategies. This limitation is especially problematic in rapidly changing environments—for instance, in crowd-based navigation—where relying on one trajectory can be inadequate [57, 58]. Moreover, the utilization of optimal transport and variational techniques has shown promise by reformulating trajectory optimization as an inference problem [58, 59]. These frameworks enable the approximation of complex solution distributions that encompass multiple local optima, thus providing the necessary flexibility for swift adaptation.

# B   Stein variational gradient descent (SVGD)

Variational inference often requires approximating an intractable posterior distribution $p(x)$ by a tractable distribution $q(x)$ from a parameterized family of distributions $\mathcal{D}$, typically through the minimization of the Kullback-Leibler (KL) divergence. Stein Variational Gradient Descent (SVGD) [10] offers a non-parametric approach that iteratively transports a set of particles $\{x_i\}_{i=1}^n$ to match the target distribution.

The core idea of SVGD is to gradually push the particles in the direction that most rapidly decreases the KL divergence. Consider an incremental, smooth transform

$$T(x) = x + \epsilon\,\phi(x),$$

where $\phi(x)$ is a smooth perturbation function and $\epsilon > 0$ is a small step size. A key theoretical result shows that the derivative of the KL divergence with respect to $\epsilon$ (evaluated at $\epsilon = 0$) is

$$\frac{d}{d\epsilon}\mathrm{KL}(q_{[T]}\|p)\bigg|_{\epsilon=0} = -\mathbb{E}_{x \sim q}\left[\mathrm{trace}\left(A_p\phi(x)\right)\right], \tag{7}$$

with the Stein operator defined as

$$A_p\phi(x) = \nabla \log p(x)\,\phi(x)^\top + \nabla_x\phi(x).$$

This result, based on Stein's identity, motivates the introduction of the kernelized Stein discrepancy (KSD) as a measure of the mismatch between $q(x)$ and $p(x)$.

By constraining $\phi$ to lie within the unit ball of a reproducing kernel Hilbert space (RKHS) associated with a positive definite kernel $k(\cdot, \cdot)$, one can derive a closed-form expression for the optimal perturbation direction:

$$\phi^*(\cdot) = \mathbb{E}_{x \sim q}\left[k(x, \cdot)\nabla \log p(x) + \nabla_x k(x, \cdot)\right]. \tag{8}$$

Here, the first term in the expectation directs particles toward regions of high probability under $p(x)$, while the second term serves as a repulsive force, ensuring that the particles remain diverse and do not collapse into a single mode.

The practical SVGD update for the particles is then given by

$$x_i \leftarrow x_i + \epsilon\,\hat{\phi}^*(x_i), \tag{9}$$

where the empirical approximation of the optimal direction is

$$\hat{\phi}^*(x) = \frac{1}{n}\sum_{j=1}^{n}\left[k(x_j, x)\nabla_{x_j}\log p(x_j) + \nabla_{x_j}k(x_j, x)\right]. \tag{10}$$

Notably, when $n = 1$ the update reduces to the standard gradient ascent used in maximum a posteriori (MAP) estimation, whereas using multiple particles allows SVGD to capture the full posterior distribution.

SVGD's formulation as a functional gradient descent in an RKHS not only avoids the need to compute Jacobians or their inverses but also scales well with large datasets by permitting mini-batch approximations of $\nabla \log p(x)$ and allows for parallel computation of each particle. This scalability, along with its ability to maintain particle diversity through the repulsive kernel term, makes SVGD an attractive and efficient method for Bayesian inference—and it forms a key component in extensions to more complex tasks.

## C  Stein variational covariance matrix adaptation evolution strategy

To overcome the reliance on gradient information in standard SVGD while retaining its powerful repulsive interactions, *Stein Variational Covariance Matrix Adaptation Evolution Strategy (SV-CMA-ES)* embeds the adaptive, natural-gradient-free CMA-ES into the multi-particle SVGD framework [26]. Conceptually, each SVGD "particle" is no longer a point but the mean of its own Gaussian search distribution. By interpreting the CMA-ES mean update as an estimate of the score function $\nabla \log p(x)$, we obtain a *gradient-free* yet highly explorative algorithm that tracks both local fitness landscapes and global diversity.

Let $n$ be the number of particles and $m$ the sub-population size per particle. Particle $i$ carries a Gaussian

$$\xi_{i,k} \sim \mathcal{N}\left(x_i, \sigma_i^2 C_i\right), \quad y_{i,k} = \frac{\xi_{i,k} - x_i}{\sigma_i}, \quad k = 1, \ldots, m,$$

and we evaluate their fitness $f(\xi_{i,k})$. By ranking these $m$ samples and selecting the top $\lambda$ as the elite sub-population, we assign recombination weights $w_{i,k}$ (non-zero only for those $\lambda$ elites, which may be positive or negative depending on their ranked fitness; see [39]).

In vanilla SVGD each particle $x_i$ moves according to

$$x_i \leftarrow x_i + \epsilon \, \mathbb{E}_{x \sim q} \big[ \nabla \log p(x) \, k(x, x_i) + \nabla_x k(x, x_i) \big].$$

In SV-CMA-ES we replace the intractable score by the *CMA-ES mean shift*

$$\Delta_j = \sum_{k=1}^{m} w_{j,k} \left( \xi_{j,k} - x_j \right) \approx \sigma_j \sum_{k=1}^{m} w_{j,k} y_{j,k}, \tag{11}$$

and approximate the expectation over all $n$ Gaussians. The resulting *SV-CMA-ES update* is

$$x_i \leftarrow x_i + \frac{\alpha_x}{n} \sum_{j=1}^{n} \Big( \Delta_j \, k(x_j, x_i) + \nabla_{x_j} k(x_j, x_i) \Big), \tag{12}$$

where $\alpha_x$ subsumes the original SVGD step-size $\epsilon$. This blend preserves SVGD's repulsion through $\nabla k$ while guiding particles toward high-fitness regions via CMA-ES's $\Delta_j$.

The self-adaptation of $\sigma_i$ and $C_i$ follows CMA-ES's mechanisms:

$$\lambda_{\text{eff}} = \Big( \sum_{k=1}^{m} w_{i,k}^2 \Big)^{-1}, \tag{13}$$

$$\bar{h}_i = \frac{\|p_{\sigma,i}\|}{\sqrt{1 - (1 - \alpha_\sigma)^{2(t+1)}}}, \tag{14}$$

$$h_{\sigma,i} = \begin{cases} 1, & \bar{h}_i < \left( 1.4 + \frac{2}{d+1} \right) \mathbb{E}\|\mathcal{N}(0, I)\|, \\ 0, & \text{otherwise}, \end{cases} \tag{15}$$

$$d(h_{\sigma,i}) = (1 - h_{\sigma,i}) \, \alpha_c \, (2 - \alpha_c) \tag{16}$$

$$\bar{w}_{i,k} = \begin{cases} w_{i,k}, & w_{i,k} \geq 0, \\ w_{i,k} \, d \, \big\| C_i^{-1/2} y_{i,k} \big\|^{-2}, & w_{i,k} < 0. \end{cases} \tag{17}$$

$$p_{\sigma,i} \leftarrow (1 - \alpha_\sigma) \, p_{\sigma,i} + \sqrt{\frac{\alpha_\sigma (2 - \alpha_\sigma)}{\lambda_{\text{eff}}}} \, C_i^{-\frac{1}{2}} \frac{\Delta_i}{\sigma_i}, \tag{18}$$

$$\sigma_i \leftarrow \sigma_i \, \exp\Big( \frac{\alpha_\sigma}{d_\sigma} \Big( \frac{\|p_{\sigma,i}\|}{\mathbb{E}\|\mathcal{N}(0,I)\|} - 1 \Big) \Big), \tag{19}$$

and

$$p_{c,i} \leftarrow (1 - \alpha_c) \, p_{c,i} + h_{\sigma,i} \sqrt{\frac{\alpha_c (2 - \alpha_c)}{\lambda_{\text{eff}}}} \frac{\Delta_i}{\sigma_i}, \tag{20}$$

$$C_i \leftarrow \Big( 1 + d(h_{\sigma,i}) - \alpha_1 - \alpha_\lambda \sum_{k=1}^{m} w_{i,k} \Big) C_i + \alpha_1 \, p_{c,i} p_{c,i}^\top + \alpha_\lambda \sum_{k=1}^{m} \bar{w}_{i,k} \, y_{i,k} y_{i,k}^\top. \tag{21}$$

**Practical considerations**

The driving-force term in (12) averages over all $n$ particles, which can shrink steps when modes are distant and cause premature step-size reduction. To address this, one may employ a *hybrid* update that uses only the particle's own CMA-ES shift for the first term and retains the full repulsion:

$$x_i \leftarrow x_i + \alpha_x \Big( \sum_{k=1}^{m} w_{i,k} (\xi_{i,k} - x_i) + \gamma(t) \frac{1}{n} \sum_{j=1}^{n} \nabla_{x_j} k(x_j, x_i) \Big), \tag{22}$$

where $\gamma(t)$ is an annealing factor (e.g. $\gamma(t) = \max(\log \frac{T}{t}, 1)$) that gradually reduces repulsion to fine-tune convergence near local optima and prevents premature convergence.

---
**Algorithm 1** Stein Variational CMA–ES
---
**Require:** kernel $k(\cdot, \cdot)$; sub-population size $m$; elite size $\lambda$; hyper-parameters $\alpha_x, \alpha_\sigma, \alpha_1, \alpha_\lambda, \alpha_c, d_\sigma$; $n$ particles; $T$ iterations

1: **Initialize:** $(x_i, \sigma_i, C_i)$ for $i = 1, \ldots, n$
2: **for** $t = 1, \ldots, T$ **do**                                    ▷ outer optimisation loop
3:     **for** $i = 1, \ldots, n$ **do**                     ▷ one search distribution per particle
    **Sample sub-population**
4:         **for** $j = 1, \ldots, m$ **do**
5:             $\xi_{i,j} \sim \mathcal{N}(x_i, \sigma_i^2 C_i)$
6:             $y_{i,j} \leftarrow (\xi_{i,j} - x_i) / \sigma_i$
7:         **end for**
    **Evaluate & accumulate statistics**
8:         compute $f(\xi_{i,j})$ and ranks $\rightarrow$ weights $w_{i,j}$
9:         $\displaystyle \Delta_i \leftarrow \sum_{k=1}^{m} w_{i,k}(\xi_{i,k} - x_i)$
10:        $\displaystyle \hat{y}_i \leftarrow \frac{1}{n} \sum_{j=1}^{n} \Big( \Delta_j \, k(x_j, x_i) + \nabla_{x_j} k(x_j, x_i) \Big)$
    **Update mean**
11:        $x_i \leftarrow x_i + \alpha_x \, \hat{y}_i$
    **Step-size (CSA)**
12:        $\displaystyle \lambda_{\text{eff}} \leftarrow \Big( \sum_{k=1}^{m} w_{i,k}^2 \Big)^{-1}$
13:        $\displaystyle p_{\sigma,i} \leftarrow (1 - \alpha_\sigma)p_{\sigma,i} + \sqrt{\frac{\alpha_\sigma(2 - \alpha_\sigma)}{\lambda_{\text{eff}}}} \, C_i^{-1/2} \, \hat{y}_i$
14:        $\displaystyle \sigma_i \leftarrow \sigma_i \, \exp\Big( \frac{\alpha_\sigma}{d_\sigma} \big( \frac{\|p_{\sigma,i}\|}{\mathbb{E}\|\mathcal{N}(0,I)\|} - 1 \big) \Big)$
    **Covariance path & matrix**
15:        compute $h_{\sigma,i}$ and $d(h_{\sigma,i})$ (CSA rule)
16:        $\displaystyle p_{c,i} \leftarrow (1 - \alpha_c)p_{c,i} + h_{\sigma,i} \sqrt{\frac{\alpha_c(2 - \alpha_c)}{\lambda_{\text{eff}}}} \, \hat{y}_i$
17:        $\displaystyle C_i \leftarrow \big( 1 + d(h_{\sigma,i}) - \alpha_1 - \alpha_\lambda \sum_{j=1}^{m} w_{i,j} \big) C_i + \alpha_1 \, p_{c,i} p_{c,i}^\top + \alpha_\lambda \sum_{j=1}^{m} \bar{w}_{i,j} \, y_{i,j} y_{i,j}^\top$
18:     **end for**
19: **end for**
---

All loops other than the outer optimization loop can be parallelized.

## D   Ergodic search and ergodic trajectory optimization

We denote the state space of a robot as $\mathcal{X} \subseteq \mathbb{X}$ and the control space as $\mathcal{U} \subseteq \mathbb{U}$, where $\mathbb{X}$ and $\mathbb{U}$ are some Banach spaces. We also denote the the continuous-time dynamics of the robot as $f(x, u) : \mathcal{X} \times \mathcal{U} \rightarrow \mathfrak{T}\mathcal{X}$, where $\mathfrak{T}\mathcal{X}$ denotes the tangent bundle of the state-space $\mathcal{X}$. Next, a robot's space trajectory $x(t) : \mathbb{R}_{\geq 0} \rightarrow \mathcal{X}$ is defined as the solution to the following initial value problem

$$x(t) = x(0) + \int_0^t f(x(\xi), u(\xi)) \, d\xi, \tag{23}$$

with initial condition $x(0) \in \mathcal{X}$, a control trajectory $u(t) : \mathbb{R}_{\geq 0} \rightarrow \mathcal{U}$. In addition, we will denote the bounded (and compact) domain (with bounds $B_i$) where the robot explores as $\mathcal{S} = [0, B_0] \times \ldots \times [0, B_{v-1}]$, where $v \leq n$. Let $g(x) : \mathcal{X} \rightarrow \mathcal{S}$ be a map that projects state space $\mathcal{X}$ to exploration space $\mathcal{S}$.

**Definition 3.** *Time-Averaged Trajectory Statistics [2]. Let $\mathcal{L}$ be the Lebesgue measure on $\mathbb{R}_{\geq 0}$ then for each $T \in \mathbb{R}_{\geq 0}$, the probability measure $\Psi_T$ on $\mathcal{S}$ that defines the time-averaged trajectory visitation statistics integrated along time $[0, T]$ is defined by*

$$\Psi_T(A) := \frac{1}{T} \mathcal{L}\big((g \circ x)^{-1}(A) \cap [0, T]\big), \tag{24}$$

*for a $A \subset \mathcal{S}$ is a Borel set.*

**Definition 4.** *Ergodicity [2, 9, 8]. A trajectory $x(t)$ is said to be ergodic with respect to a Borel probability measure $\pi$ on $\mathcal{S}$ if $\Psi_T$ converges weakly to $\pi$ as $T \to \infty$, formally,*

$$\lim_{T \to \infty} \int_{\mathcal{S}} \phi(s)\, d\Psi_T(s) = \int_{\mathcal{S}} \phi(s)\, d\pi(s), \quad \forall \phi \in \mathcal{C}(\mathcal{S}). \tag{25}$$

*The trajectory statistics measure can be simplified as an integral of delta functions, where*

$$\int_{\mathcal{S}} \phi(s)\, d\Psi_T(s) = \frac{1}{T} \int_0^T \phi(g \circ x(t)))\, dt. \tag{26}$$

Intuitively, a trajectory is ergodic with respect to the measure $\pi$ if it explores $A$ in a manner that is commensurate with $\pi$ as $T \to \infty$.

**Definition 5.** *Ergodic Cost Function [2]. Given a probability measure $\pi$ on $\mathcal{S}$. A $\pi$-ergodic cost function is defined as $\mathcal{E}_\pi : \tau(\mathcal{X}) \to \mathbb{R}$ such that for any infinite trajectory $x(t) : \mathbb{R}_{\geq 0} \to \mathcal{X}$, if*

$$\lim_{T \to \infty} \mathcal{E}_\pi(x|_{[0,T]}) \to 0 \tag{27}$$

*then $x(t)$ is ergodic, where*

$$\tau := \bigcup_{T > 0} \mathcal{C}^0([0, T], \mathcal{X}), \tag{28}$$

as the set of all continuous trajectories defined on arbitrary finite time intervals. Throughout this article we stick to $\mathbb{X} = \mathbb{R}^n$ and $\mathbb{U} = \mathbb{R}^m$ and, we use the spectral methods and construct a metric in the Fourier space to define the ergodic metric for trajectory optimization [2, 8, 52, 1].

**Definition 6.** *Spectral Ergodic Cost Function. Let $\pi$ be a probability measure on $\mathcal{S}$. Let $\mathcal{K}^v \subset \mathbb{N}^v$ be the set of all integer $k$ fundamental frequencies that define the cosine Fourier basis function*

$$F_k(w) = \frac{1}{h_k} \prod_{i=0}^{v-1} \cos\left(\frac{w_i k_i \pi}{L_i}\right) \tag{29}$$

*where $h_k$ is a normalizing factor (see [8]). For a finite trajectory $x(t) : [0, T] \to \mathcal{X}$, let $\Psi_T$ be the measure defined in Eq. (24). The spectral ergodic cost function is defined as*

$$\mathcal{E}_\pi(x) = \sum_{k \in \mathcal{K}^v} \Lambda_k \left(\Psi_T^k - \lambda^k\right)^2 \tag{30}$$

$$= \sum_{k \in \mathcal{K}^v} \Lambda_k \left(\frac{1}{T} \int_0^T F_k(g \circ x(t)) dt - \int_{\mathcal{S}} F_k(s) d\pi(s)\right)^2$$

*where $\Psi_T^k$ and $\mu^k$ are the $k^{th}$ Fourier decomposition modes of $\Psi_T$ and $\mu$, respectively (using Eq. (26)), and $\Lambda_k = (1 + \|k\|_2)^{-\frac{v+1}{2}}$ is a weight coefficient that places higher importance on lower-frequency modes.*

In particular, the spectral ergodic cost function defined for a probability measure $\pi$ forms a metric [8].

Alternatively, one can use the kernel MMD to construct a ergodic metric for trajectories on any generalized domains [60].

We now can formulate an (ergodic) trajectory optimization using the ergodic metric as

$$\begin{aligned} \underset{\substack{x(t) \in \tau(\mathcal{X}) \\ u(t) \in \mathcal{U}, \forall t \in [0,T]}}{\text{minimize}} \quad & \mathcal{E}_\pi(x(t)) \tag{31} \\ \text{subject to} \quad & h_1(x) = 0, \quad \forall t \in [0, T] \\ & h_2(x) \leq 0, \quad \forall t \in [0, T] \end{aligned}$$

where $h_1$ are the quality constraints e.g., $\dot{x} = f(x, u)$ and $h_2$ are the inequality constraints.

# E Stein variational ergodic search

## E.1 Likelihood of ergodicity and stein variations

Stein Variational Ergodic Search (SVES) [2] recasts the trajectory optimization problem as a (variational) inference problem. They begin by introducing a binary indicator function

$$O : \tau(\mathcal{X}) \to \{0, 1\},$$

which flags whether a given trajectory satisfies a desired optimality condition. Let $\pi$ be a probability measure defined on the exploration domain $\mathcal{S}$. The likelihood that a trajectory $x$ exhibits the desired ergodic behavior through an ergodic cost function $\mathcal{E}_\pi(x)$ is given by

$$p(O \mid x) = \exp\Big(-\mu\, \mathcal{E}_\pi(x)\Big),$$

where $\mu > 0$ is a scaling parameter that governs the influence of the ergodic cost.

By combining this likelihood with a prior $p(x)$ over the space of trajectories via Bayes' rule, one can obtain the posterior $p(x \mid O)$. This posterior is approximated using Stein Variational Gradient Descent (SVGD). In this framework, the update direction that minimizes the Kullback-Leibler divergence (from equation 8) is given by

$$\phi^*(\cdot) = \mathbb{E}_{x \sim q}\Big[k(x, \cdot)\left(\nabla_x \log p(x) - \mu\,\nabla_x \mathcal{E}_\pi(x)\right) + \nabla_x k(x, \cdot)\Big] \tag{32}$$

where $k(x, \cdot)$ is a positive definite kernel on the space of trajectories. This vector field steers each trajectory sample toward regions of lower ergodic cost while ensuring diversity across samples.

## E.2 Ergodic stein variational trajectory optimization

For computational tractability, we discretize trajectories. Let a trajectory be represented as

$$\mathbf{x} = [x_0, x_1, \dots, x_{T-1}],$$

with $x_t \in \mathcal{X}$. An empirical distribution over $N$ discrete trajectory samples is given by

$$\hat{q} = \frac{1}{N} \sum_{i=1}^{N} \delta_{\mathbf{x}^i}.$$

Typically one assumes a Gaussian prior of the form

$$p(\mathbf{x}) = \mathcal{N}(\hat{\mathbf{x}}, \sigma^2),$$

where $\hat{x}$ is typically chosen as an interpolation between specified boundary conditions.

Constraints on the trajectory can be integrated into the cost likelihood function in the following form

$$\mathcal{J}_\pi(\mathbf{x}) = \mathcal{E}_\pi(\mathbf{x}) + \rho(\mathbf{x}) + c_1 h_1(\mathbf{x})^2 + c_2 \max(0, h_2(\mathbf{x})) \tag{33}$$

where $c_1, c_2$ are (positive) Lagrange multipliers that form an inner product with equality and inequality functions $h_1, h_2$, and $\rho : \tau(\mathcal{X}) \to \mathbb{R}$ is any additional penalty terms on the trajectory $\mathbf{x}$. [3] Now, the ergodic Stein variational update step using $\mathcal{J}_\pi$ is given by

$$\phi_r^*(\cdot) = \frac{1}{N} \sum_{i=1}^{N} [k(\mathbf{x}_r^i, \cdot)(\nabla_\mathbf{x} \log p(\mathbf{x}_r^i) - \mu\nabla_\mathbf{x} \mathcal{J}_\pi(\mathbf{x}_r^i)) + \nabla_\mathbf{x} k(\mathbf{x}_r^i, \cdot)]. \tag{34}$$

using the kernel a $k(\mathbf{x}, \cdot)$. This procedure exploits the repulsive characteristics of the kernel to prevent sample (or mode) collapse, thereby ensuring that the resulting trajectories remain diverse well-dispersed and effective for exploration. It is crucial to note that maintaining diversity and preventing mode collapse fundamentally depends on selecting an appropriate and effective kernel. The algorithm outlining this procedure is as follows:

---

[3]One can use augmented Lagrangian or any interior point method to guarantee constraint satisfiable (although in practice this may not be required in most cases). Alternatively one may choose integrate constraints as described in [61]

---

**Algorithm 2** Stein Variational Ergodic Trajectory Opt.

---

1: **input:** measure $\pi$, domain $\mathcal{S}$, map $g \colon \mathcal{X} \to \mathcal{S}$, cost $\mathcal{J}_\pi$, prior $p(x)$, kernel $k(x, \cdot)$, step size $\epsilon$, iteration $r = 0$, initial trajectory samples $\{x_i^0\}_{i=1}^N$, termination condition $\gamma$
2: **while** Some convergence criterion **do**
3:      **for all** samples $i$ in parallel **do**
4:          $x_{r+1}^i \leftarrow x_r^i + \epsilon\,\phi_r^*(x_r^i)$
5:      **end for**
6:      $r \leftarrow r + 1$
7: **end while**
8: **return:** $\{x_i^r\}_{i=1}^N$, $\arg\max_i \exp\!\big(-\mu\,\mathcal{J}_\pi(x_i^r)\big)$

---

### E.3 Ergodic stein variational control

This approach can easily be extended to the control domain by optimizing over sequences of control inputs rather than trajectories directly. Let the control sequence be represented as

$$\mathbf{u} = [u_0, u_1, \ldots, u_{T-1}],$$

and let the the system evolves according to the discrete-time dynamics

$$x_{t+1} = F(x_t, u_t), \quad \text{given } x_0.$$

We define an empirical distribution over control sequences by

$$\hat{q} = \frac{1}{N} \sum_{i=1}^N \delta_{\mathbf{u}^i},$$

and assume a prior $p(u)$ on the control sequences. With a kernel $k(u, \cdot)$ defined on the control space, the SVGD update for controls is expressed as

$$\phi_r^*(\cdot) = \frac{1}{N} \sum_{i=1}^N [k(\mathbf{u}_r^i, \cdot)(\nabla_{\mathbf{u}} \log p(\mathbf{u}_r^i) - \mu \nabla_{\mathbf{u}} \mathcal{J}_\pi(\mathbf{x}_r^i | \mathbf{u}_r^i, x_0)) + \nabla_{\mathbf{u}} k(\mathbf{u}_r^i, \cdot)], \quad (35)$$

where $\mathcal{J}_\pi(\mathbf{x}_r^i | \mathbf{u}_r^i, x_0)$ represents the ergodic (with constraints) cost incurred by the trajectory generated under the control sequence $u_i$ starting from the initial state $x_0$. This control formulation is well-suited for online applications, where the first control action is executed and the sequence is re-optimized at each time step to adapt to new information.

The algorithm summarizes the approach for control optimization within a model-predictive control framework is as follows:

---

**Algorithm 3** Stein Variational Ergodic Control

---

1: **input:** initial state $x_0$, time horizon $T$, measure $\pi$, domain $W$, map $g \colon \mathcal{X} \to \mathcal{S}$, prior $p(u)$, kernel $k(u, \cdot)$, step size $\epsilon$, prior control samples $\{u_i^0\}_{i=1}^N$, termination condition $\gamma$
2: $r \leftarrow 0$
3: **while** Some convergence criterion **do**
4:      **for all** samples $i$ in parallel **do**
5:          $u_{r+1}^i \leftarrow u_r^i + \epsilon\,\phi_r^*(u_r^i)$
6:      **end for**
7:      $r \leftarrow r + 1$
8: **end while**
9: **return:** $\{u_i^r\}_{i=1}^N$, $i^* = \arg\max_i \exp\!\big(-\mu\,\mathcal{J}_\pi(u_i^r)\big)$
10: apply $u_0^*$ to robot
11: /* shift controls */
12: **for all** samples $i$ in parallel **do**
13:      $u_{i,0:T-2} \leftarrow u_{i,1:T-1}$
14: **end for**
15: /* sample state and return to input */

---

# F  Convergence of SVGD in the Ergodic Search Setting

The work in [2] shows that the general convergence results for SVGD from [30] remain valid in the ergodic search setting when using an RBF kernel. We now generalize this convergence result to any differentiable kernel.

Consider a discretized ergodic search on a bounded state space $\mathcal{X} \subset \mathbb{R}^n$ and a normalized workspace $\mathcal{W} = [0,1]^v$ with $T$ time points. Without loss of generality we set $\mathcal{X} = [0,1]^n$, and thus SVGD acts on the space $\mathcal{X}^T = [0,1]^{Tn}$. We fix a mapping $g \colon \mathcal{X} \to \mathcal{X}$. Moreover, we work in the population limit of infinitely many initial samples from the prior.

Let $\mathcal{E}_\pi$ be the spectral ergodic cost function with respect to a measure $\pi$ on $\mathcal{W}$, and let $p$ be the prior on $\mathcal{X}^T$. Then the posterior

$$\eta := p(x \mid O) \; \propto \; p(O \mid x)\, p(x)$$

can be written in the form

$$\eta \sim \exp\big(-\mu\, \mathcal{E}_\pi(x) + \log p(x)\big). \tag{27}$$

Define the potential function

$$V(x) = \mu\, \mathcal{E}_\pi(x) - \log p(x), \tag{28}$$

as is standard in SVGD.

Furthermore, let $k \colon \mathcal{X}^T \times \mathcal{X}^T \to \mathbb{R}$ be a positive definite kernel with associated RKHS $\mathcal{H}$. Convergence of SVGD is then characterized by the kernel Stein discrepancy (KSD). In particular, for any measure $q$, its discrepancy to $\eta$ is

$$\mathrm{KSD}_\eta(q) := \|\phi^*(q)\|_{\mathcal{H}^T}^2, \tag{29}$$

where $\phi_q^* \in \mathcal{H}^T$ (noting $\mathcal{X}^T \subset \mathbb{R}^{Tn}$) is given by

$$\phi^*(q) = \mathbb{E}_{x \sim q}\big[k(x,\cdot)\big(\nabla_x \log p(x) - \mu\, \nabla_x \mathcal{E}_\pi(x)\big) \;+\; \nabla_x k(x,\cdot)\big]. \tag{30}$$

In particular, $\mathrm{KSD}_\eta(q)$ coincides with the norm of the SVGD update gradient $\phi_r^*$ in (32). Here we define SVGD in the population limit in the same way as the finite-sample case. The population gradient at iteration $r$ is $\phi_r^* := \phi^*(q_r)$, where $q_r$ is the pushforward of $q_{r-1}$ under the map $U_r \colon \mathcal{X}^T \to \mathcal{X}^T$,

$$U_r(x) := x + \epsilon\, \phi_{r-1}^*(x), \tag{31}$$

with step size $\epsilon > 0$ and initial distribution $q_0 = p$.

Moreover, the convergence results in [30] rest on three assumptions on the kernel $k$, the potential $V$, and the updates $U_r$. Specifically, there exist constants $B, C, M > 0$ such that:

  (A1)  $\|k(x,\cdot)\|_{\mathcal{H}} \leq B$ and $\|\nabla_x k(x,\cdot)\|_{\mathcal{H}} \leq B$.
  (A2)  The Hessian $H_V$ of $V$ in (28) is well-defined with operator norm $\|H_V\|_{\mathrm{op}} \leq M$.
  (A3)  For all $r$, $\mathrm{KSD}_\eta(q_r) \leq C$.

Now we can formally state the generalized convergence guarantee from [2] as follows:

**Theorem 2.** *Let $\mathcal{X} = [0,1]^n$, let $\eta = p(x \mid O)$ be the target with potential $V$ from (28), and assume $p$ is smooth on $\mathcal{X}^T$. Let $k$ be a positive kernel on $\mathcal{X}^T$ with RKHS $\mathcal{H}$. Then there exists a step size $\epsilon < S$, where $S$ depends on $B, C, M$, such that*

$$\frac{1}{r} \sum_{i=1}^r \mathrm{KSD}_\eta(q_i) \;\leq\; \frac{\mathrm{KL}(p\|\eta)}{c_\epsilon\, r}, \tag{32}$$

*where $c_\epsilon$ is a constant depending on $\epsilon$, $M$, and $B$.*

*Proof.* This result is a special case of [30, Corollary 6], and in order to prove this result, we must show that assumptions $(A1), (A2), (A3)$ above hold. First, $(A1)$ holds as long as the kernel $k$ is differentiable, and $\mathcal{X}^T$ is a bounded domain. Next, we note that for a discrete path $\mathbf{x} = [x_0, \dots, x_{T-1}]$, the spectral ergodic cost fucntion has the form

$$\mathcal{E}_\pi(\mathbf{x}) = \sum_{k \in \mathcal{K}^v} \Lambda_k \left( \frac{1}{T} \sum_{t=0}^{T-1} F_k(g(x_t)) - \int_{\mathcal{W}} F_k(w) d\pi(w) \right)^2, \tag{36}$$

which is smooth since $F_k$ from Eq. (29) is smooth. Because the prior $p$ is also smooth, the potential $V$ is smooth, and the Hessian is well-defined. Furthermore, the Hessian is bounded since $\mathcal{X}^T$ is a bounded domain, so $(A2)$ is satisfied. Finally, since $(A1)$ and $(A2)$ is satisfied, it suffices to show that

$$\sup_r \int_{\mathcal{X}^T} \|\mathbf{x}\| \, dq_r(\mathbf{x}) < \infty, \tag{37}$$

to show $(A3)$, from the discussion in [30, Section 5]. However, this is satisfied since $q_r$ is a probability measure and $\mathcal{W}^T$ is a bounded domain, so $(A3)$ is satisfied. $\qquad\square$

# G   Kernels

## G.1   Radial basis function (RBF) kernel

The Radial Basis Function (RBF) kernel [62], also known as the Gaussian kernel, is one of the most widely used kernels in machine learning and other applications due to its strong theoretical properties and practical performance. Given two inputs $x, y \in \mathbb{R}^d$ (In the case of our trajectories, $\mathbf{x} = [x_1, \ldots, x_T]$ where $x_i \in \mathbb{R}^n$, we flatten them into a vector in $\mathbb{R}^{Tn}$), the RBF kernel is defined as

$$k_{\mathrm{RBF}}(x, x') \;=\; \exp\!\left(-\frac{\|x - x'\|^2}{2h^2}\right), \tag{38}$$

where $h > 0$ is the kernel bandwidth, and is chosen using the median heuristic. This choice of kernel corresponds to an inner product in an infinite-dimensional reproducing kernel Hilbert space (RKHS), making it a universal approximator under mild conditions on $h$.

## G.2   Markov RBF kernel

Given two state trajectories

$$\mathbf{x} = [x_1, \ldots, x_T], \quad \mathbf{y} = [y_1, \ldots, y_T],$$

we define a Markov kernel by decomposing similarity along the Markov chain:

$$k_{\mathrm{M}}(\mathbf{x}, \mathbf{y}) \;=\; \sum_t k(x_t, y_t) + \sum_{(t,t')\in\mathcal{G}} k(x_t, y_{t'})$$

where each $k$ is a some (positive definite) kernel and $\mathcal{G}$ is a graph of connected points separated by time i.e., $\mathcal{G}$ chosen to encode the Markov-structure along time (typically the nearest-neighbour edges $t \leftrightarrow t+1$).

**Choice of the temporal graph $\mathcal{G}$.**   For a sparse Markov kernel one usually takes $\mathcal{G}_{\mathrm{nn}} = \{(t, t+1)\}_{t=1}^{T-1}$, so that only nearest-neighbour time indices interact. This nearest-neighbour choice was the one employed in the experiment described in section 7.2.

In the complete-graph variant used in the experiments in sections 7.1 & 7.3.

$$\mathcal{G}_{\mathrm{complete}} \;=\; \big\{\, (t, s) \,\big|\, 1 \le t \le T, \; 1 \le s \le T, \; t \neq s \big\},$$

which connects every ordered pair of distinct time indices. Substituting this into the definition gives

$$k_{\mathrm{M}}^{\mathrm{complete}}(\mathbf{x}, \mathbf{y}) \;=\; \sum_{t=1}^{T} \sum_{\substack{s=1 \\ s\neq t}}^{T} k(x_t, y_s),$$

and dividing the double sum by $T^2$ simply rescales the kernel without affecting positive-definiteness or SVGD updates.

A typical choice for the base kernel is the radial basis function (RBF) with bandwidth $h$ set by, the median heuristic in our experiments. This construction yields an expressive yet efficient measure of trajectory similarity that honors the one-step dependence characteristic of Markov processes.

### G.3 Global alignment kernel

Let $\mathbf{x} = [x_1, \ldots, x_n]$ and $\mathbf{y} = [y_1, \ldots, y_m]$ be two trajectories (of possibly different lengths) in $\mathbb{R}^d$. Traditional vector kernels (e.g., Gaussian or polynomial) cannot directly handle variable-length sequences nor account for temporal distortions. Dynamic alignment methods such as Dynamic Time Warping (DTW) (see Appendix G.4) overcome this by finding an optimal alignment, but do not yield positive-definite kernels in general.

The *Global Alignment* (GA) kernel [63] addresses these issues by *summing* over *all* possible alignments between $x$ and $y$. Formally, let

$$
A(\mathbf{x}, \mathbf{y}) = \left\{ \pi = ((i_1, j_1), \ldots, (i_L, j_L)) : \begin{array}{l} 1 = i_1 \leq i_2 \leq \cdots \leq i_L = n, \\ 1 = j_1 \leq j_2 \leq \cdots \leq j_L = m, \\ (i_{k+1} - i_k) \in \{0, 1\}, \ (j_{k+1} - j_k) \in \{0, 1\} \end{array} \right\}
$$

be the set of all monotonic alignments between the indices of $\mathbf{x}$ and $\mathbf{y}$. Given a local positive-definite kernel (often called static kernel)

$$
k : \mathcal{X} \times \mathcal{X} \longrightarrow \mathbb{R},
$$

the GA kernel is defined as

$$
k^{ga}(\mathbf{x}, \mathbf{y}) = \sum_{\pi \in A(x,y)} \prod_{(i,j) \in \pi} k(x_i, y_j).
$$

Intuitively, $K(x, y)$ aggregates the similarity contributions of every alignment, acting as a "soft-max" over DTW-like scores. In our experiments we chose the RBF kernel as the static kernel.

**Positive definiteness.** Under mild conditions on $k$ (for instance, if both $k$ and $k/(1+k)$ are positive definite), $K$ is itself a Mercer kernel.

**Efficient computation.** Although $|A(\mathbf{x}, \mathbf{y})|$ grows exponentially, $k^{ga}(\mathbf{x}, \mathbf{y})$ admits a simple dynamic programming algorithm in $O(nmd)$ time. Define

$$
M_{i,j} = \begin{cases} 1, & i = j = 0, \\ 0, & i = 0 < j \text{ or } j = 0 < i, \\ k(x_i, y_j) \left( M_{i-1,j} + M_{i,j-1} + M_{i-1,j-1} \right), & i, j \geq 1. \end{cases}
$$

Then

$$
K(x, y) = M_{n,m}.
$$

This quadratic-time computation matches the complexity of standard DTW while incorporating richer alignment information. And this can be further reduced to a wall-clock time complexity of $O(md)$ (without loss of generality on $n$ and $m$) on a GPU that can accommodate the required number of threads.

### G.4 Kernelized dynamic time warping

Let $\mathbf{x} = [x_1, \ldots, x_n]$ and $\mathbf{y} = [y_1, \ldots, y_m]$ be two trajectories (of possibly different lengths) in $\mathbb{R}^d$. Traditional vector kernels cannot handle temporal misalignments and variable lengths inherent in time-series comparison. Dynamic Time Warping (DTW) addresses this by computing an optimal alignment path $\pi^*$ that minimizes the cumulative local distance:

$$
\delta_{\text{DTW}}(\mathbf{x}, \mathbf{y}) = \min_{\pi \in A(\mathbf{x}, \mathbf{y})} \sum_{i=1}^{|\pi|} \left\| x_{\pi_1(i)} - y_{\pi_2(i)} \right\|^2.
$$

**Kernelization of DTW (KDTW).** [64] To turn DTW into a Mercer kernel many but very similar ideas have been used, we will state 2 such ideas. One approach is to exponentiate the optimal cost, yielding the *dynamic time-alignment kernel*:

$$
k^{\text{DTW1}}(\mathbf{x}, \mathbf{y}) = \exp\left( \frac{1}{2h^2} \left( - \min_{\pi \in A(\mathbf{x}, \mathbf{y})} \sum_{i=1}^{|\pi|} \left\| x_{\pi_1(i)} - y_{\pi_2(i)} \right\|^2 \right) \right).
$$

An alternative, inspired by convolution kernels [65], is to exponentiate local distances first and then maximize over alignments:

$$k^{\mathrm{DTW2}}(\mathbf{x}, \mathbf{y}) = \max_{\pi \in A(\mathbf{x},\mathbf{y})} \frac{1}{|\pi|} \sum_{i=1}^{|\pi|} \exp\left(-\frac{1}{2h^2}\left\|x_{\pi_1(i)} - y_{\pi_2(i)}\right\|^2\right).$$

Neither construction guarantees differentiability.

We use $k^{\mathrm{DTW1}}$ in our experiments and $h$ is chosen using the median heuristic at every iteration of SVGD or SV-CMA-ES. We approximated the gradient of the $k^{\mathrm{DTW1}}$ kernel using a centered finite-difference scheme in our SVGD experiments. Although the KDTW kernels are not differentiable in the classical sense, we observed no adverse effects in our experiments. Nevertheless, these theoretical issues warrant careful attention in future work.

**Computational complexity.** Both $k^{\mathrm{DTW1}}$ and $k^{\mathrm{DTW2}}$ admit $O(nmd)$ dynamic-programming algorithms, replacing the usual $\min$ operations of DTW with exponentiation and $\max$ (or further $\min$) steps, while preserving quadratic time complexity. And this can be further reduced to a wall-clock time complexity of $O(md)$ (without loss of generality on $n$ and $m$) on a GPU that can accommodate the required number of threads.

### G.5 Signature kernel

**Definition 7.** *Signature (transform) of a path (or trajectory) [20]. Let $V$ be a Banach space and define the tensor algebra of formal power series*

$$T((V)) = \prod_{k \geq 0} V^{\otimes k},$$

*which is equipped with the natural operations induced by the tensor product and has the unit element $(1, 0, 0, \dots)$.*

*Now, consider a continuous path $x : I \to V$ on a compact interval $I \subset \mathbb{R}$ with finite $p$-variation (for some $p < 2$). For any subinterval $[s, t] \subset I$, the* signature *of $x$ over $[s, t]$ is defined as the sequence*

$$\varphi(x)_{s,t} = \left(1, \int_{s < u_1 < t} dx(u_1), \dots, \int_{s < u_1 < \cdots < u_k < t} dx(u_1) \otimes \cdots \otimes dx(u_k), \dots\right) \in T((V)). \tag{39}$$

*This collection of iterated integrals captures the essential features of the path and is invariant under reparameterization.*

Recall, the signature of a trajectory is injective (for all non-tree-like paths) and, remains unchanged under any reparametrization. This invariance effectively filters out the complex, infinite-dimensional group of symmetries. Moreover, the collection of linear functionals defined on the signature not only forms an algebra under pointwise multiplication but also has the power to separate points [66]. Consequently, by the *Stone–Weierstrass* theorem, for any compact set $\mathbf{C}$ of continuous paths with bounded variation, the set of such linear functionals is dense in the space of continuous real-valued functions on $\mathbf{C}$ [67]. These properties together establish the signature as an ideal feature map for representing our trajectories in ergodic search.

For a path $x$ of finite $p$-variation with $p > 1$ the signature terms decay factorially [67]

$$\left\|\int_{s < u_1 < \cdots < u_k < t} dx(u_1) \otimes \cdots \otimes dx(u_k)\right\|_{V^{\otimes k}} \leq \frac{\|x\|_{p,[s,t]}^k}{k!}$$

where $\|x\|_{p,[s,t]}$ is the $p$-variation of the path on $[s, t]$. Therefore, one can approximate signature kernel reasonably well with the *truncated signature transform* of level $L \in \mathbb{N}$,

$$\varphi_{[s,t]}^L(x) = \left(1, \int_{s < u_1 < t} dx(u_1), \dots, \int_{s < u_1 < \cdots < u_L < t} dx(u_1) \otimes \cdots \otimes dx(u_L)\right) \in \bigoplus_{k=0}^{\infty} V^{\otimes k}, \tag{40}$$

where $\oplus$ denotes the direct sum, and [21] and [22] provide efficient means to compute the truncated signature using Dynamic Programming and approximate it using Random Fourier Features, respectively.

**Definition 8.** *Signature Kernel Let $I = [u, u']$ and $J = [v, v']$ be two compact intervals, and consider paths $x \in \mathcal{C}^1(I, V)$ and $y \in \mathcal{C}^1(J, V)$, where $V$ is a Banach space. The signature kernel $k^{sig}(x, y) : I \times J \to \mathbb{R}$ is defined by*

$$k_{s,t}^{sig}(x, y) = \langle \varphi(x)_s, \ \varphi(y)_t \rangle, \tag{41}$$

*where $\varphi(x)_s$ and $\varphi(y)_t$ denote the signatures of $x$ and $y$ over the intervals $[u, s]$ and $[v, t]$, respectively.*

Utilizing the *kernel trick* introduced in [20], one can efficiently compute the full (untruncated) signature kernel.

**Theorem 3.** *Let $I = [u, u']$ and $J = [v, v']$ be compact intervals, and let $x \in \mathcal{C}^1(I, V)$ and $y \in \mathcal{C}^1(J, V)$. Then the signature kernel $k_{x,y}$ satisfies the following linear, second-order hyperbolic partial differential equation:*

$$\frac{\partial^2 k_{s,t}^{sig}}{\partial s \, \partial t}(x, y) = \langle \dot{x}(s), \ \dot{y}(t) \rangle_V \, k_{s,t}(x, y),$$

*subject to the boundary conditions*

$$k_{u,t}^{sig}(x, y) = 1 \quad \forall t \in J, \qquad k_{s,v}^{sig}(x, y) = 1 \quad \forall s \in I.$$

The resulting PDE formulation of the signature kernel enables its efficient computation using any standard numerical scheme, such as finite difference, finite element, or other suitable methods. An important point to note is the assumptions on Theorem 3, will be satisfied for any reasonable discretizations of the trajectories and the domain over which the PDE is solved.

One must take extra cation if they wish to use the signature kernel in SV-CMA-ES, since the trajectories produced in initial few iterations of SV-CMA-ES may not be very well behaved, can produce very rough trajectories, breaking the assumptions of Theorem 3.

### G.6 Median heuristic

The median heuristic [27] sets the bandwidth

$$h = \frac{\text{med}\big(\{\|x_i - x_j\|_2\}_{i<j}\big)}{\log N},$$

using the empirical median of all pairwise distances among particles (divided by $\log N$) to provide a robust, data-driven scale. We adopt

$$h = \frac{\text{med}\,(\{\mathbf{x}\})}{\log N},$$

which yields slightly improved performance (in terms of computational speed) in our experiments. Additionally, we employed the median heuristic for bandwidth selection because it is straightforward, computationally light, and requires no manual tuning, making our method more accessible to users. We recommend referring to [27] for more (theoretical) guarantees, advantages and details on the median heuristic.

## H Measuring trajectory diversity

One can interpret the kernel $k(\mathbf{x}, \mathbf{y})$ as a similarity measure between two trajectories $\mathbf{x}$ and $\mathbf{y}$. In particular, for a normalized kernel function (i.e., $k(x, x) = 1$), $k(\mathbf{x}, \mathbf{y}) \to 1$ as $\mathbf{x} \to \mathbf{y}$, $k(\mathbf{x}, \mathbf{y}) \to 0$ as $\|\mathbf{x} - \mathbf{y}\| \to \infty$. Given a collection of trajectories $\{\mathbf{x}^i\}_{i=1}^N$, one may quantify their diversity by forming the Gram matrix $K \in \mathbb{R}^{N \times N}$, $K_{ij} = k(\mathbf{x}^i, \mathbf{x}^j)$, $K_{ii} = 1$, and computing its determinant. Notably, $\det(K) \to 0$ if $\mathbf{x}^i \to \mathbf{x}^j$ for any $i \neq j$, $\det(K) \to 1$ as $\|\mathbf{x}^i - \mathbf{x}^j\| \to \infty$ for all $i \neq j$, thereby providing a principled measure of particle diversity.

Throughout the experiments we shall use the following kernel for measuring diversity,

$$k^{\text{Frechet}}(x, y) = \exp\left(-\frac{d_{\text{continuous Frechet}}(x, y)^2}{2h^2}\right) \tag{42}$$

where $d_{\text{continuous Frechet}}(\cdot, \cdot)$ is the *(Continuous) Frechet distance* (see Appendix I) and $h = 0.1$. We denote the Gram matrix of this kernel with $K_f$.

One may choose any finite-valued kernel to measure diversity, we chose $k^{\text{Frechet}}$ simply due to the fact that it has not been used in experiments to benchmark the signature kernel.

## I   Fréchet distance

The Fréchet distance [68] is a classical similarity measure between trajectories that respects both spatial geometry and the ordering of points along each trajectory. Intuitively, it is the minimum leash-length necessary for a person and a dog to walk along two continuous paths, each at its own varying speed, without backtracking.

**Definition 9** (Continuous Fréchet Distance). *Let $f, g \colon [0,1] \to (\mathcal{X}, d)$ be continuous curves in a metric space. A* reparameterization *is a continuous, nondecreasing, surjection $\alpha \colon [0,1] \to [0,1]$ satisfying*

$$\alpha(0) = 0, \quad \alpha(1) = 1,$$

*(and similarly for $\beta$). The* continuous Fréchet distance *is*

$$d_{\text{Continuous Frechet}}(f, g) = \inf_{\substack{\alpha, \beta \\ \text{reparam.}}} \max_{t \in [0,1]} d\big(f(\alpha(t)), g(\beta(t))\big).$$

In our case with discretized trajectories in $\mathbb{R}^d$,

$$\mathbf{x} = [x_1, \ldots, x_n], \quad \mathbf{y} = [y_1, \ldots, y_m],$$

we use the discrete Fréchet distance.

**Definition 10** (Discrete Fréchet Distance). *A coupling of $\mathbf{x}$ and $\mathbf{y}$ is a pair of index sequences $\sigma = (\sigma(1), \ldots, \sigma(K))$ and $\tau = (\tau(1), \ldots, \tau(K))$ such that $\sigma(1) = 1$, $\sigma(K) = n$, $\tau(1) = 1$, $\tau(K) = m$, and each of $\sigma, \tau$ is non-decreasing with increments at most 1. The discrete Fréchet distance is*

$$d_{\text{Frechet}}(\mathbf{x}, \mathbf{y}) = \min_{\sigma, \tau \text{ couplings}} \max_{k=1,\ldots,K} \| x_{\sigma(k)} - y_{\tau(k)} \|.$$

**Computational complexity**   The continuous Fréchet distance can be computed in

$$O\big(d\, n\, m\, \log(nm)\big),$$

time [69, 70]. Here—and unlike most of the literature that treats $d$ as a hidden constant—$d$ varies in our setting and so is made explicit in the $O(\cdot)$ notation.

**Properties**

- *Metric Properties.* The continuous Fréchet distance is only a pseudometric on the space of parameterized curves—distinct parameterizations of the same geometric trace satisfy $d_{\text{Continuous Frechet}}(f, g) = 0$—but it induces a true metric on the quotient of curves under reparameterization. In contrast, the discrete Fréchet distance $d_{\text{Frechet}}$ on point-sequences satisfies all four metric axioms (non-negativity, symmetry, identity of indiscernibles, and the triangle inequality) and therefore is a bona fide metric on the set of discrete trajectories

- *Stability.* If two curves $f, f'$ and $g, g'$ satisfy $\|f - f'\|_\infty \le \varepsilon$ and $\|g - g'\|_\infty \le \varepsilon$, then

$$\big|d(f, g) - d(f', g')\big| \le \varepsilon.$$

- *Order-sensitivity.* Unlike the Hausdorff distance, the Fréchet distance respects the natural parameterization of trajectories, making it especially powerful for comparing time-indexed data.

Because of these advantages, the Fréchet distance has become a cornerstone in shape matching, motion analysis, and trajectory mining.

## J   Additional results

### J.1   Multiscale constrained exploration

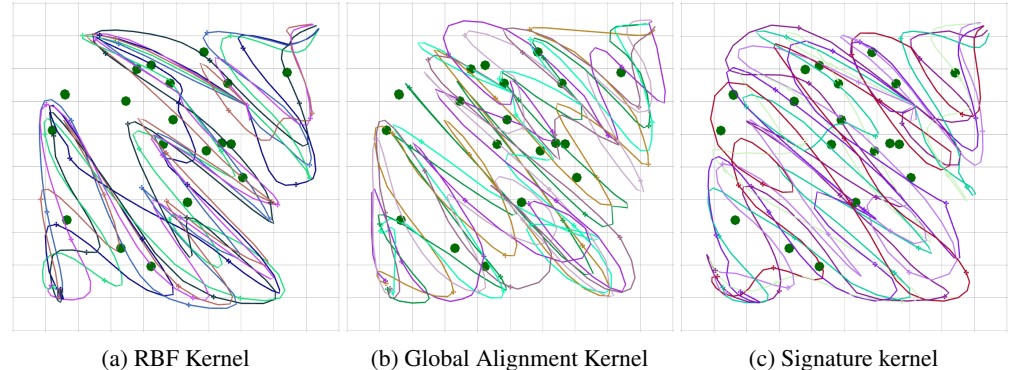

(a) RBF Kernel       (b) Global Alignment Kernel       (c) Signature kernel

Figure 9: Trajectories generated using different kernels in the multiscale constrained exploration experiment comparing kernels in section 7.1.1.

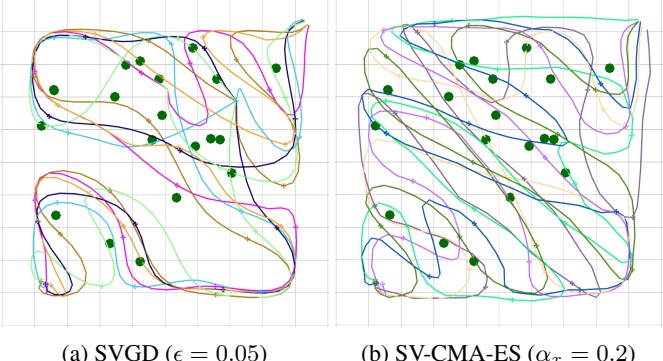

(a) SVGD ($\epsilon = 0.05$)       (b) SV-CMA-ES ($\alpha_x = 0.2$)

Figure 10: Trajectories generated using SVGD and SV-CMA-ES in the multiscale constrained exploration experiment comparing the SVGD and SV-CMA-ES in section 7.1.2.

Table 2: Aggregated metrics across 10 priors for each kernel for the Multiscale constrained exploration

| Kernel | Metric | Mean | StdDev | Min | Max |
|---|---|---|---|---|---|
| RBF | Best trajectory cost | 0.4203 | 0.0297 | 0.3742 | 0.4833 |
| | Average trajectory cost | 0.5152 | 0.0399 | 0.4190 | 0.5666 |
| | Mean ergodic cost | 0.1816 | 0.0194 | 0.1308 | 0.1979 |
| | Trajectory diversity | 2.2052 | 0.7413 | 1.0172 | 3.5636 |
| Signature Kernel | Best trajectory cost | 0.4145 | 0.0564 | 0.3423 | 0.5479 |
| | Average trajectory cost | 0.4916 | 0.0541 | 0.4065 | 0.5995 |
| | Mean ergodic cost | 0.1677 | 0.0234 | 0.1295 | 0.2120 |
| | Trajectory diversity | 3.3854 | 1.2080 | 1.8504 | 5.4397 |
| Kernelized DTW | Best trajectory cost | 0.4249 | 0.0228 | 0.3925 | 0.4771 |
| | Average trajectory cost | 0.5275 | 0.0457 | 0.4320 | 0.6185 |
| | Mean ergodic cost | 0.1865 | 0.0229 | 0.1368 | 0.2215 |
| | Trajectory diversity | 1.9260 | 0.6717 | 0.8498 | 3.3909 |
| Markov RBF | Best trajectory cost | 0.5270 | 0.0496 | 0.4421 | 0.6267 |
| | Average trajectory cost | 0.5346 | 0.0488 | 0.4501 | 0.6326 |
| | Mean ergodic cost | 0.1762 | 0.0187 | 0.1519 | 0.2122 |
| | Trajectory diversity | 0.00005 | 0.00002 | 0.00003 | 0.00009 |
| Global Alignment Kernel | Best trajectory cost | 0.4767 | 0.0626 | 0.3763 | 0.6247 |
| | Average trajectory cost | 0.5732 | 0.0599 | 0.4912 | 0.6928 |
| | Mean ergodic cost | 0.1740 | 0.0299 | 0.1200 | 0.2274 |
| | Trajectory diversity | 2.8324 | 1.3845 | 0.7104 | 5.5709 |

## J.2 3D Coverage using the Crazyflie drone

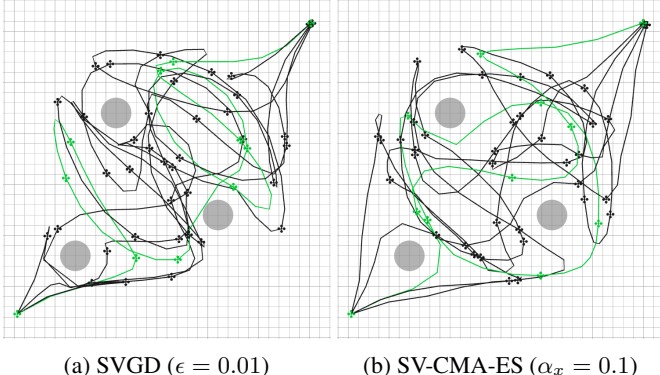

(a) SVGD ($\epsilon = 0.01$)  (b) SV-CMA-ES ($\alpha_x = 0.1$)

Figure 11: Top down projection of trajectories generated using SVGD and SV-CMA-ES in the 3D coverage using the Crazyflie drone experiment comparing the SVGD and SV-CMA-ES in section 7.2.2.

Table 3: Aggregated metrics across 10 priors for each kernel for the 3D coverage using the crazyflie drone

| Kernel | Metric | Mean | StdDev | Min | Max |
|---|---|---|---|---|---|
| RBF | Best trajectory cost | 1.1923 | 0.0415 | 1.1105 | 1.2498 |
| | Average trajectory cost | 1.3216 | 0.0213 | 1.2884 | 1.3700 |
| | Mean ergodic cost | 0.7118 | 0.0172 | 0.6868 | 0.7380 |
| | Trajectory diversity | 7.0293 | 1.0004 | 5.2130 | 8.4017 |
| Signature Kernel | Best trajectory cost | 1.0071 | 0.0323 | 0.9503 | 1.0509 |
| | Average trajectory cost | 1.1283 | 0.0171 | 1.0923 | 1.1596 |
| | Mean ergodic cost | 0.5574 | 0.0143 | 0.5319 | 0.5798 |
| | Trajectory diversity | 10.3330 | 1.0336 | 8.4339 | 11.4770 |
| Kernelized DTW | Best trajectory cost | 1.0140 | 0.0247 | 0.9698 | 1.0588 |
| | Average trajectory cost | 1.1413 | 0.0183 | 1.1026 | 1.1695 |
| | Mean ergodic cost | 0.5702 | 0.0121 | 0.5412 | 0.5866 |
| | Trajectory diversity | 8.9372 | 1.6642 | 6.0877 | 12.2700 |
| Markov RBF | Best trajectory cost | 1.3248 | 0.0440 | 1.2547 | 1.4031 |
| | Average trajectory cost | 1.4934 | 0.0310 | 1.4592 | 1.5560 |
| | Mean ergodic cost | 0.8445 | 0.0257 | 0.8037 | 0.8860 |
| | Trajectory diversity | 5.1746 | 1.3442 | 2.6743 | 7.4214 |
| Global Alignment Kernel | Best trajectory cost | 1.1251 | 0.0347 | 1.0638 | 1.1850 |
| | Average trajectory cost | 1.2507 | 0.0209 | 1.2183 | 1.2833 |
| | Mean ergodic cost | 0.6489 | 0.0126 | 0.6245 | 0.6741 |
| | Trajectory diversity | 7.3018 | 1.7350 | 4.4249 | 9.9842 |

## J.3 Stein variational model-predictive control

Table 4: Aggregated metrics from Table 1 across all 10 priors.

| Kernel | Metric | Mean | StdDev | Min | Max |
|---|---|---|---|---|---|
| RBF | Total best trajectory cost | 382.347 | 14.278 | 377.832 | 422.982 |
| | Total average trajectory cost | 574.443 | 35.744 | 563.140 | 676.174 |
| | Total ergodic cost of best trajectory | 268.319 | 19.699 | 262.090 | 324.385 |
| | Total trajectory diversity | 49.986 | 4.655 | 48.514 | 63.233 |
| Signature | Total best trajectory cost | 360.060 | 23.569 | 340.256 | 420.901 |
| Kernel | Total average trajectory cost | 592.608 | 46.984 | 552.196 | 719.126 |
| | Total ergodic cost of best trajectory | 244.615 | 24.056 | 225.835 | 309.706 |
| | Total trajectory diversity | 566.583 | 153.891 | 360.379 | 925.329 |
| Markov | Total best trajectory cost | 476.244 | 23.475 | 468.820 | 543.055 |
| RBF | Total average trajectory cost | 775.057 | 34.687 | 764.088 | 873.777 |
| | Total ergodic cost of best trajectory | 328.503 | 27.263 | 319.882 | 406.095 |
| | Total trajectory diversity | 3795.151 | 679.803 | 2050.583 | 4080.356 |
| Kernelized | Total best trajectory cost | 364.486 | 13.236 | 347.587 | 383.277 |
| DTW | Total average trajectory cost | 561.385 | 14.056 | 542.033 | 581.960 |
| | Total ergodic cost of best trajectory | 256.216 | 13.246 | 240.409 | 276.713 |
| | Total trajectory diversity | 68.739 | 12.963 | 55.162 | 98.367 |

## K   Implementational and experimental deatils

In this section we provide the full Implementational and experimental details.

### K.1   Implementational details

**Elite sample selection and log-rank weights for SV-CMA-ES**   In our experiments we rank the $m$ samples by selecting the top

$$\lambda \;=\; \left\lfloor \tfrac{m}{5} \right\rfloor$$

as the elite sub-population, we assign *positive log-rank* recombination weights

$$w_{i,k} \;=\; \frac{\log(\lambda + 0.5) - \log r_k}{\displaystyle\sum_{j=1}^{\lambda}\big[\log(\lambda + 0.5) - \log j\big]}, \qquad r_k = 1, \ldots, \lambda,$$

and set $w_{i,k} = 0$ for $k > \lambda$. ("Active" negative weights [39] are *disabled* in the present implementation.)

**Enhancing the signature kernel's performance**   A common enhancement is to first embed each trajectory into a (possibly infinite-dimensional) feature space via a static feature map $\varphi_{\text{static}} : \mathbb{R}^d \longrightarrow \mathcal{H}$, associated, for example, with a radial basis or Matérn kernel. Concretely, one defines

$$\varphi_{\text{static}}(\mathbf{x}) \;:=\; \big(\varphi_{\text{static}}(x(i))\big)_{i=1}^{T}.$$

Empirical studies demonstrate that this pre-lifting step substantially improves the performance of the signature kernel. Therefore, the ergodic Stein variational update step in the state space using the

signature kernel and a static kernel is given by

$$\phi_r^*(\cdot) = \frac{1}{N} \sum_{i=1}^{N} [k^{sig}(\varphi_{static}(\mathbf{x}_r^i), \varphi_{static}(\cdot))(\nabla_{\mathbf{x}} \log p(\mathbf{x}_r^i) -$$

$$\mu \nabla_{\mathbf{x}} \mathcal{J}_\pi(\mathbf{x}_r^i)) + \nabla_{\mathbf{x}} k^{sig}(\varphi_{static}(\mathbf{x})_r^i, \varphi_{static}(\cdot))]. \quad (43)$$

An analogous update applies in the control space. Throughout this paper, we emploied the feature map corresponding to the radial basis kernel (RBF) (see Appendix G).

**Hardware used**    All experiments were run on the a computer with:

1. **Operating System:** Ubuntu 20.04.6 LTS
2. **CPU:** AMD Ryzen 9 5900HS
3. **RAM:** 16 GB
4. **GPU:** NVIDIA GeForce RTX 3060
5. **CUDA Version:** 12.2

**Assets used**    The codebases provided by the following works [2, 22, 20] and the GitHub repository https://github.com/khdlr/softdtw_jax were either directly incorporated into our experiments or proved essential as implementation references.

## K.2    Experimental details

### K.2.1    Multiscale constrained exploration

We consider the two-dimensional spatial domain $\mathcal{S} = [0, 100] \times [0, 100]$ m, equipped with the uniform measure $\pi$. To improve numerical conditioning, we introduce the affine mapping $g \colon \mathcal{S} \to [0, 1] \times [0, 1]$, $g(x) = \frac{x}{100}$. [4] Trajectories of dimension $n = 2$ and length $T = 100$ with a time step of $\Delta t = 1$ s are generated via Algorithm described in sections 3.1 & E.2. A maximum $k_{\max} = 8$ of basis functions per dimension is employed for constructing the ergodic metric (see Appendix D Definition 6). The cost functional is specified as

$$\mathcal{J}_\pi(\mathbf{x}) = \mathcal{E}_\pi(\mathbf{x}) + 1 \, c_{\mathcal{S}}(\mathbf{x}) + \sum_{t=0}^{T-1} 15 \left\| x_{t+1} - x_t \right\|_2^2$$

$$+ 0.1 \left\| x_0 - x_{\text{init}} \right\|_2^2 + 0.1 \left\| x_T - x_{\text{final}} \right\|_2^2 + 0.01 \, c_{\text{obs}}(\mathbf{x}), \quad (44)$$

where the obstacle penalty is defined by $c_{\text{obs}}(\mathbf{x}) = \max\left(0, \left\| x - x_c \right\|_2 - r\right)$, with $x_c$ and $r$ denoting the obstacle center and radius, respectively. The initial and final states, $\mathbf{x}_{\text{init}}$ and $\mathbf{x}_{\text{final}}$, are prescribed on the boundary of $\mathcal{S}$.

---

[4]This normalization does not affect the optimization, since the ergodic metric is defined in the Fourier spectral domain on any periodic domain.

Table 5: Full Hyperparameter overview of SV-CMA-ES in the multiscale constrained exploration example

| Hyperparameter | Value |
| --- | --- |
| $m$ | 32 |
| $\lambda$ | 6 |
| $\sigma_i^{(0)}$ | 0.1 (initial self adaption rate) |
| $\alpha_\sigma$ | $(\lambda_{\text{eff}} + 2)/(M + \lambda_{\text{eff}} + 5)$, where $M$ is problem dimension, i.e, for a trajectory with $T$ time steps and in $d$ dimensions, $M = Td$ |
| $d_\sigma$ | $1 + 2\max\left(0, \sqrt{\frac{\lambda_{\text{eff}} - 1}{M+1}} - 1\right) + \alpha_\sigma$ |
| $\alpha_c$ | $(4 + \lambda_{\text{eff}}/M)/(M + 4 + 2\lambda_{\text{eff}}/M)$ |
| $\alpha_1$ | $2/((M + 1.3)^2 + \lambda_{\text{eff}})$ |
| $\alpha_\lambda$ | $\min\left(1 - \alpha_1,\ 2(\lambda_{\text{eff}} - 2 + 1/\lambda_{\text{eff}})/((M + 2)^2 + 2\lambda_{\text{eff}}/2)\right)$ |
| $k(\cdot, \cdot)$ | RBF kernel |

### K.2.2 3D Coverage using the Crazyflie drone

We consider a three-dimensional domain $\mathcal{S} = [0, 3] \times [0, 3] \times [0.5, 1.5]$, equipped with the uniform measure $\pi$. Trajectories are computed via Algorithm described in section 3.1 & E.2, using a cost functional analogous to that of the multiscale forest example:

$$\mathcal{J}_\pi(\mathbf{x}) = \mathcal{E}_\pi(\mathbf{x}) + 0.1\, c_{\mathcal{S}}(\mathbf{x}) + \sum_{t=0}^{T-1} 15 \left\| x_{t+1} - x_t \right\|_2^2$$
$$+ 1\left\| x_0 - x_{\text{init}} \right\|_2^2 + 0.1\left\| x_T - x_{\text{final}} \right\|_2^2 + 0.01\, c_{\text{obs}}(\mathbf{x}). \quad (45)$$

The trajectory length is set to $T = 150$ with a time step of $\Delta t = 0.1$s. A maximum $k_{\max} = 8$ of basis functions per dimension is employed for constructing the ergodic metric (see Appendix D Definition 6).

Table 6: Full Hyperparameter overview of SV-CMA-ES

| Hyperparameter | Value |
| --- | --- |
| $m$ | 32 |
| $\lambda$ | 6 |
| $\sigma_i^{(0)}$ | 0.2 (initial self adaption rate) |
| $\alpha_\sigma$ | $(\lambda_{\text{eff}} + 2)/(M + \lambda_{\text{eff}} + 5)$, where $M$ is problem dimension, i.e, for a trajectory with $T$ time steps and in $d$ dimensions, $M = Td$ |
| $d_\sigma$ | $1 + 2\max\left(0, \sqrt{\frac{\lambda_{\text{eff}} - 1}{M+1}} - 1\right) + \alpha_\sigma$ |
| $\alpha_c$ | $(4 + \lambda_{\text{eff}}/M)/(M + 4 + 2\lambda_{\text{eff}}/M)$ |
| $\alpha_1$ | $2/((M + 1.3)^2 + \lambda_{\text{eff}})$ |
| $\alpha_\lambda$ | $\min\left(1 - \alpha_1,\ 2(\lambda_{\text{eff}} - 2 + 1/\lambda_{\text{eff}})/((M + 2)^2 + 2\lambda_{\text{eff}}/2)\right)$ |
| $k(\cdot, \cdot)$ | RBF kernel |

### K.2.3 Stein variational model-predictive control

We consider coverage over a quad-modal target distribution

$$\pi(x) = \frac{1}{4} \sum_{i=1}^{4} \mathcal{N}\big(\mu_i, \ \sigma_i^2 I_2\big),$$

comprising four Gaussian components

$$\eta_1 = (0.2, \ 0.2), \qquad \sigma_1 = \frac{1}{\sqrt{300}},$$

$$\eta_2 = (0.85, \ 0.85), \quad \sigma_2 = \frac{1}{\sqrt{300}},$$

$$\eta_3 = (0.23, \ 0.75), \quad \sigma_3 = \frac{1}{\sqrt{300}},$$

$$\eta_4 = (0.75, \ 0.2), \qquad \sigma_4 = \frac{1}{\sqrt{300}},$$

on the unit-square domain $\mathcal{S} = [0, 1] \times [0, 1]$. The agent obeys single-integrator dynamics $x_{t+1} = x_t + \Delta t \, u_t$, with $\Delta t = 0.1$. Trajectory planning is carried out over a horizon of $T = 20$ time steps, using $N = 20$ control-sequence samples per iteration. Each planning loop is executed for up to 200 iterations.

A maximum $k_{\mathrm{max}} = 10$ of basis functions per dimension is employed for constructing the ergodic metric (see Appendix D Definition 6). Ten obstacles are placed uniformly at random in $\mathcal{S}$; each obstacle moves at the same rate $\Delta t$, with velocity directions sampled from $\mathcal{N}(0, \sigma^2)$ where $\sigma = 0.01$.

The cost functional for a control sequence $\mathbf{u} = (u_0, \dots, u_{T-1})$ is defined as

$$\mathcal{J}_\pi(\mathbf{u}) = \mathcal{E}_\pi(\mathbf{u}) + c_{\mathcal{S}}(\mathbf{x}) + 0.01 \, \|\mathbf{u}\|_2^2 + 0.001 \sum_{t=0} \|x_{t+1} - x_t\|_2^2 + 100 \, c_{\mathrm{obs}}(\mathbf{x}), \qquad (46)$$

with scaling parameter $\mu = 10$. The initial control prior is taken as $p(\mathbf{u}) = \mathcal{N}\big(0, \ 0.01 \, I\big)$.

