# OpenReview forum: "Diversifying Parallel Ergodic Search: A Signature Kernel Evolution Strategy"
_NeurIPS.cc/2025/Conference — NeurIPS 2025 poster_

### Official Review · Reviewer_eVaC · 2025-06-20

**Clarity:** 2
**Significance:** 2
**Originality:** 3
**Rating:** 4
**Confidence:** 2

**Summary:**

This paper proposes a new method of ergodic search for robots based on the SVES framework, and it aims to improve the diversity of the generated trajectories. The key innovation is to replace the traditional RBF kernel with the Signature Kernel, which improves the kernel's sensitivity to differences in paths. The paper also introduces a PDE formulation of computing the signature kernel that allows parallel computation. The experiments show that with the signature kernel, SVES has better state coverage and convergence results.

**Questions:**

Typo:
You assume $h_2(x) \leq 0$ in Line 122, but the last term in Eq. 3 is $\max(0, h_2(x))$, which equals to 0.


Question:
1. Do you have evaluation metrics to measure the trajectory smoothness? I am concerned about whether the signature kernel will result in sudden accelerations or sharp turns.
2. Could you explain in more detail how the signature kernel differs from the RBF kernel when it comes to comparing two similar trajectories?

**Ethical Concerns:**

["NO or VERY MINOR ethics concerns only"]

**Final Justification:**

Issues resolved:
1. Explaining the difference between the RBF and signature kernel in differentiating two trajectories: The authors provided a toy example, and I believe the authors should further include a figure to better visualize it.
2. Explaining how their method affects the trajectory smoothness: I find that the authors specify the hyperparameters controlling the smoothness in Eq. 42, Appendix J, which is the complete loss function they use. I think the authors should move it to the main text (Section 3), because Section 3 omits part of their loss function.

Issues unresolved:
1. Compare SVES with the signature kernel with the kernel-free baselines: The authors did not include the recent robotic exploration methods without kernels. Although the authors claimed that they addressed it in the Introduction part, I think the necessity of using the SVES framework in robotic exploration tasks isn't well established yet. Therefore, I think it's better to directly including the kernel-free methods as the baselines, so that it will be more reasonable to replace the traditional RL framework by their SVES framework.
2. Including experiments in other environments: The authors only use one environment in their experiments. The authors claimed that their main contribution is to "demonstrate diversification within SVES", not to propose a new framework because the signature kernel already exists in existing works (See [11][17]][18] in their paper). Although they provided references showing that SVES framework adapts to other environments, I'm still confused whether the experiment within only one environment is enough to "demonstrate diversification".

Therefore, I raise my rating to 4, but I think the authors should address the unresolved issues in their revised manuscript.

**Limitations:**

The authors addressed the limitations and potential negative societal impact of their work.

**Paper Formatting Concerns:**

No paper formatting concerns.

**Quality:**

3

**Strengths And Weaknesses:**

Strength:
1. The paper contains both qualitative comparisons and quantitative results of trajectories generated using the RBF kernel and the Signature kernel, illustrating that the Signature kernel helps cover the state space.
2. Using signature kernels for ergodic exploration is an innovative idea, whereas previously signature kernels were applied to trajectory optimization in robotics, as in “Path Signatures for Diversity in Probabilistic Trajectory Optimisation.”
3. The paper extends the theoretical framework in the prior work Stein Variational Ergodic Search (SVES).


Weakness:
1. The reason why the signature kernel can improve the state coverage is still unclear. In Line 78, the author claims "The signature kernel naturally discriminates between **subtly different paths**, thus promoting diversity." Can you visualize how the signature kernel helps discriminate the local differences between two similar paths, or provide a detailed explanation about why the RBF kernel fails to do so while the signature kernel can?
2. The experiments only compare SVES with the signature kernel with SVES based on other kernels. However, there have been plenty of kernel-free methods for the robot exploration problem. For example, Burda et al. (2018) “Exploration by Random Network Distillation” introduces **intrinsic rewards** to improve the trajectories' diversity. Also, the **curiosity-driven** approaches in "Aggressive Quadrotor Flight Using Curiosity-Driven Reinforcement Learning" and "Curiosity-Driven Reinforcement Learning based Low-Level Flight Control" also help improve the state coverage during training. I am confused about whether the SVES framework is necessary for the ergodic search problem.
3. The experiments are only conducted in the 3D Coverage Drone Experiment, which follows a similar setting in Lee et al. (2024) "Stein Variational Ergodic Search". I am concerned about the applicability of the signature kernel in other robotic tasks.

---

> ### Author Rebuttal · Authors · 2025-07-30
>
> ## Note of Gratitude
>
> We thank the reviewer for the response and the insights provided that can help strengthen the paper.
>
> # 1. Do you have evaluation metrics to measure the trajectory smoothness? I am concerned about whether the signature kernel will result in sudden accelerations or sharp turns.
> The strong repulsive force provided by the signature kernel may at first seem like a recipe for producing rough and non-smooth trajectories, as producing rough trajectories is one way to boost diversity. However, we found that this did not have an adverse effect on the smoothness. We include a smoothness penalty in the cost/loss function $\mathcal J$ to enforce trajectory regularity (see Sections 2 \& 3 and Appendix J and the resulting trajectories are visualized in Figure 1, Section 7 and Appendix I) and SVES guarantees convergence to a locally optimal trajectory with mild conditions on the chosen kernel (that signature kernel satisfies). The proof of convergence is provided in Appendix A of [1]. So, the trajectories should converge to a set of smooth trajectories provided that there is a smoothness penalty in the cost/loss function.
>
> After the rebuttal, an ablation study that tunes the trajectory smoothness and compares the performance of the kernels with respect to the trajectory smoothness will be provided along with the proof of convergence.
>
> # 2. Could you explain in more detail how the signature kernel differs from the RBF kernel when it comes to comparing two similar trajectories?
>
> We provide a brief mathematical reasoning for why the signature kernel makes a good kernel, it is discussed in lines 154-158 \& 768-775 (the reviewer may also refer to references from [11,14,15,19,61,62] from our paper). A short and oversimplified answer is that the signature transform is injective  on all non-tree-like paths, meaning no two such paths share the same signature. But this is not the case for the RBF kernel. Consider paths:
> 1. $p_1(t) = (x_1(t),y_1(t)) = (t, 0) \quad t \in [0, 2\pi]$
> 2. $p_2(t) = (x_2(t),y_2(t)) = (t, sin(t)) \quad t \in [0, 2\pi]$
> 3. $p_3(t) = (x_3(t),y_3(t)) = (t, sin(t + \pi)) \quad t \in [0, 2\pi]$
>
>
> It is easy to see that $k^{RBF}(p_1,p_2) = k^{RBF}(p_1,p_3)$. But this not the case with the signature kernel due the the signature transform being invective up to “tree‑like” equivalence.
>
> # Weakness
> ## Weakness 2:
> This is briefly discussed in Section 1 of our paper. In short, classical ergodic planners focus on steering a single trajectory (or a centrally controlled fleet) to match
>     a target spatial distribution. In contrast, our SVES‐diversification methods generate an set of distinct or diverse ergodic trajectories in parallel, leveraging repulsive interactions or signature‐based seeding to ensure both high coverage and trajectory diversity in a unified optimization. Refer to Section 1 \& 2 of [1] and Section 1 and Appendix A of our paper for a more formal take on this. The reviewer can also choose to refer the response provided to reviewer KSg3.
> ## Weakness 3:
> Because our primary goal is to demonstrate diversification within SVES, we intentionally replicated the ergodic search experiments from the original Stein Variational Ergodic Search paper. Throughout the paper, we also also include references to prior successes of the signature kernel in robotics applications, for example:
> 1. reference [11] from our paper used the signature kernel to produce a diverse set of solutions in motion planning tasks such as manipulation.
> 2. reference [17] from our paper used the signature transform for Human Action Recognition.
> 3. reference [18] from our paper used the signature kernel for trajectory following/stabilization.
>
> In addition, the reviewer may also refer to [2], where the authors used the signature transform in data driven control.
>
> # Typo:
> ## Clarification of the Constrained vs. Penalized Formulation
>
> In the main text we first posed the constrained optimization problem as
> $$ {\text{minimize}} \quad \mathcal{E}_\pi(x(t)), \qquad \text{subject to} \quad h_1(x) = 0, \quad h_2(x) \le 0, \quad \forall t \in [0, T].  \qquad \quad (1)$$
>
>
> There is no typo in Eq 3 of our paper; instead, we switch to an unconstrained,
> *penalty‑augmented* version of (1) by adding a hinge‐loss
> term that *penalizes* any violation of the inequality constraint.  Concretely,
> we define
>
> $$\mathcal J_\pi(x)
>         =\mathcal E_\pi(x)
>         +\rho(x)
>         +c_1\,h_1(x)^2
>         +c_2\max (0,h_2(x)). \qquad \quad (2)$$
>
> ## Why this is correct and not a typo:
> 1. The hinge function $\max(0,h_{2}(x))$ is $0$ whenever
>         $h_{2}(x)\le0$, i.e. the original constraint is satisfied.
> 2. If $h_{2}(x)$ becomes positive (constraint violation),
>         $\max(0,h_{2}(x))=h_{2}(x)>0$, so the penalty term
>         $c_{2}\,h_{2}(x)$ grows linearly, discouraging infeasible
>         solutions.
> 3. Thus (2) is a standard “soft‐constraint” or
>         “penalty” reformulation of the hard constraint in
>         (1), not a mistake in the sign or definition.
>
> # References
>
> [1] Darrick Lee, Cameron Lerch, Fabio Ramos, and Ian Abraham. Stein Variational Ergodic Search. arXiv
> preprint arXiv:2406.11767, 2024. https://arxiv.org/abs/2406.11767
>
> [2] Anna Scampicchio and Melanie N. Zeilinger. On the role of the signature transform in nonlinear systems
> and data-driven control. arXiv preprint arXiv:2409.05685, 2025. https://arxiv.org/abs/2409.05685

---

> > ### Comment · Reviewer_eVaC · 2025-08-04
> >
> > Thanks for your clarification! The authors resolved my concerns about the trajectory smoothness and the difference between the RBF kernel and the signature kernel. I updated my rating accordingly.
> > I recommend that the authors do the following two revisions in their final manuscript:
> > 1. I recommend that the authors move Eq. 42 in Appendix J to Eq. 3 in Section 3, so that the cost function $\mathcal J$ will be clear. According to the authors' response to my question "Do you have evaluation metrics to measure the trajectory smoothness", the authors include a smoothness penalty $15||x_{t+1} - x_t||^2$ in the cost/loss function $\mathcal J$ to enforce trajectory regularity in Eq. 42, Line 891 in Appendix J. However, in Eq. 3, the smoothness penalty is included in the "additional penalties" $\rho$, and $\rho$ is not defined in Section 3.
> > 2. I recommend that the authors include your toy example $p_1, p_2, p_3$ in your appendix, so that the difference between the RBF and signature kernel can be illustrated. You can also include a figure to illustrate the toy example if possible.

---

### Official Review · Reviewer_cUiH · 2025-07-01

**Clarity:** 3
**Significance:** 3
**Originality:** 2
**Rating:** 4
**Confidence:** 4

**Summary:**

Efficiently exploring a bounded domain within a finite amount of time is a challenging and central problem in robotics. Ergodic exploration converts exploration into a coverage problem, optimizing the time-average spatial distribution of the robot against an expected information measure. While this can handle multi-modal landscapes, in practice, the non-convex nature of the objective makes this difficult. Stein Variational Ergodic Search (SVES) was proposed to mitigate this problem, leveraging Stein Variational Gradient Descent (SVGD), which approximates the distributions of trajectories in a particle-based fashion. However, the choice of kernel in SVGD can have a huge impact on the diversity of the trajectories considered. In this paper, the authors propose to use the signature kernel, which can better discriminate between trajectories and respects its Markovian structure. In addition, the authors propose to use Stein Variational Covariance Matrix Adaptation Evolution Strategy (SV-CMA-ES) instead of SVGD, which is gradient-free and better take advantage of GPU parallelism. This is motivated by both the fact that gradients can be expensive to compute and that search-based strategies have shown empirical benefits over gradient-based methods. To estimate the gradient of the signature kernel, they propose to use Simultaneous Perturbation Stochastic Approximation (SPSA), which is also a gradient-free method that evaluates a black-box function via random perturbations. They evaluate SV-CMA-ES across two benchmark systems and five choices of kernels (including the signature kernel). Their results show that trajectories produced via the signature kernel are indeed more diverse, lower cost, and slightly more ergodic than other choices of kernels. They demonstrate improved convergence of SV-CMA-ES over SVGD on both benchmark systems. Finally, they use the signature kernel in SV-MPC, a model predictive control algorithm which leverages SVGD to perform the updates. They show that the signature kernel improves yields trajectories with more diversity while remaining low cost. They also found that the Markov RBF kernel failed to converge to meaningful solutions.

**Questions:**

- How do the computational costs of each choice of kernel compare? What about the cost for SV-CMA-ES over SVGD? I know this depends a lot on the choice of hardware, but some numbers would help give a ball-park idea.

**Ethical Concerns:**

["NO or VERY MINOR ethics concerns only"]

**Limitations:**

- The authors provide a number of limitations, mostly centered around computational cost. They also talk about how the repulsive force with the signature kernel can be almost too strong, encouraging poor choices, such a collisions into obstacles. They also provide some remedies for these issues and possible future directions.

**Quality:**

3

**Strengths And Weaknesses:**

Strengths:
- Efficient exploration of a predefined region in a timely manner is an important and core problem in robotics. This paper combines existing tools in a novel way to solve this problem, with clear benefits in terms of trajectory cost and diversity on two simulated platforms. They also show their choice of kernel can be used in an MPC loop, with clear benefits over other choices of kernels.
- They perform sufficient ablations that show the concrete benefits of each of their design choices. In particular, they evaluate four different choices of baseline kernels compared with their choice of signature kernel. They also explore the convergence of SV-CMA-ES and compare it to SVGD, showing the benefits of this choice of optimizer.
- Figure 1 is nicely placed and immediately introduces the reader to the empirical benefits of their approach, with a clear improvement in the diversity of trajectories. Figure 2 also makes it clear which kernel works better for this application.
- The appendix is contains most of the additional information necessary to understand the paper, making it fairly self-contained.
- In terms of writing, I enjoyed how the main paper focused on the high-level points, relegating the details to the appendix for the interested reader. I also liked how the core questions of the evaluations were outlined in the introduction of Section 7.

Weaknesses:
- Using only two benchmark systems on similar tasks is a great starting point, but the evaluations could be more extensive. Exploring systems with different choices of dimensionality, types of dynamics, and obstacle types and configurations would strengthen this paper.
- Figure 3 and 5 make it difficult to clearly see how each method compares. I know there's a table in the appendix, but the visualization of the data could be much clearer. For instance, pairing up by metric rather than kernel could help make the differences distinguishable. Or you could show the distributions with box or violin plots, instead of averages, dividing each metric into a separate sub-panel. While not critical, it would make it much easier to understand the results.
- In general, there isn't enough discussion of the results or conclusions drawn. In particular, Figures 4 and 6 provide the comparison in convergence of SV-CMA-ES versus SVGD without any discussion or interpretation of the results in the text. There are also a lot of curves in the plot, with substantial overlap, making it hard to interpret. The paper would be stronger if the discussion of these results was elaborated.
- It would be great to have a little exposition on the benchmark problems used to evaluate the method. While I like relegating a lot of the details to the appendix, the considered problems do need to be introduced more in the main paper.
- Model predictive control (MPC) is discussed and used in this paper, without really being defined, at least in the main paper. Since it is an important part of the evaluation, I would suggest providing at least a couple sentences describing how it works and adding citations to SV-MPC in the main paper.
- Similarly, it would be great if the median heuristic would be defined, or at least given a citation. I know it's a fairly standard practice, but for completeness, it would be good to include.

---

> ### Author Rebuttal · Authors · 2025-07-30
>
> # 1. How do the computational costs of each choice of kernel compare? What about the cost for SV-CMA-ES over SVGD? I know this depends a lot on the choice of hardware, but some numbers would help give a ball-park idea.}
>
> We thank the reviewer for pointing out the lack of computational benchmarks in our original submission. We agree that providing concrete estimates of computational costs for different kernel choices, as well as comparing the runtime complexity of SV-CMA-ES versus SVGD, would help clarify the practical implications of our method. We provide the following analysis in response.
>
> ## Theoretical Computational Cost Comparisons
> ### Kernels Computational cost
> We provided the computational costs for some kernels in Section 5 and Appendix F.
> 1. The time complexity of the RBF kernel is \(O(Td)\), but in practice can be as low as $O(d)$ on modern hardware with T threads.
>
> 2. The time complexity of MRBF is $O\bigl(|\mathcal G| \cdot O(\mathrm{RBF})\bigr)$.
>
> 3. The signature kernel, the global alignment kernel and the kernelized DTW can be computed in $O(Td)$ with modern parallelized hardware.
>
> ### SV-CMA-ES vs SVGD Computational cost
> The original Stein Variational Gradient Descent (SVGD) paper (reference [10] from our paper) identifies the gradient of the log‑density as its primary computational bottleneck. Comparing the costs of SVGD and SV‑CMA‑ES is therefore nontrivial, since both methods require evaluating gradients of the kernel and of the log‑density at each update. The key difference lies in the cost of obtaining the log‑density gradient itself. When that gradient is easy to compute or available analytically, SVGD will generally be the more efficient choice. By contrast, SV‑CMA‑ES truly shines when the log‑density is nondifferentiable or expensive to evaluate.
>
> In practice, computing the log‑density gradient typically entails two passes through the density function (a forward and a backward pass). Evolution‐strategy (ES) updates, however, are highly parallelizable on GPUs. The ES updates require multiple parallel forward passes of the log density, and computing the weighted recombination step for each particle can all be batched on modern hardware (with say Cholesky decomposition). Thus, even though SV‑CMA‑ES performs more sampling “work,” its overhead beyond objective evaluations is often quite small. If the log density is non‑differentiable or costly to backpropagate, SV‑CMA‑ES can actually deliver faster wall‑clock convergence by avoiding gradient calls altogether.  Moreover SV-CMA-ES can have faster convergence rates when compared to SVGD with well-tuned hyperparameter(see reference [25] from our paper).
>
> ## Experimental Benchmarks
>
> In the following, we report mean run-times (over 10 independent runs) for SVGD and SV‑CMA‑ES on a common hardware platform, and discuss implementation caveats.
>
> ### Experimental Setup
> The SVGD implementation was fully JIT‑compiled, whereas SV‑CMA‑ES was only partially JIT‑compiled. No optimizations were applied to the signature kernel, so absolute timings may differ under a more optimized implementation. All experiments were run using the following hardware:
>
> 1. **CPU**: AMD Ryzen 7 7800X3D (8 cores @ 4.2 GHz)
> 2. **Memory**: 32 GB (@ 5600 MT/s)
> 3.  **GPU**: NVIDIA Geforce RTX 5060 Ti
>
>
> ### Results
> | Experiments               | Multiscale constrained exploration (Section 7.1) | 3D Coverage (Section 7.2) | SV‑MPC (Section 7.3)         |
> |---------------------------|--------------------------------------------------|----------------------------|-----------------------------|
> | No. of iterations         | 3000                                             | 1500                       | 100                         |
> | Problem dimensions | $T=100, \ d=2$                            |  $T=50, \ d=3$           | $T=20, \ d=2$ |
> | N particles               |                                                  |                            |                             |
> |                           |                                                  | **SVGD with RBF kernel**   |                             |
> | 5                         | 2.83 ± 0.08 seconds                              | 1.45 ± 0.17 seconds        | 0.092 ± 0.0034 seconds      |
> | 10                        | 2.82 ± 0.12 seconds                              | 1.45 ± 0.11 seconds        | 0.093 ± 0.0034 seconds      |
> | 15                        | 2.90 ± 0.14 seconds                              | 1.40 ± 0.09 seconds        | 0.095 ± 0.0078 seconds      |
> | 20                        | 3.11 ± 0.17 seconds                              | 1.46 ± 0.10 seconds        | 0.090 ± 0.0043 seconds      |
> | 25                        | 2.94 ± 0.15 seconds                              | 1.44 ± 0.13 seconds        | 0.094 ± 0.0055 seconds      |
> |                           |                                                  | **SVGD with Signature kernel** |                          |
> | 5                         | 13.83 ± 0.84 seconds                             | 6.32 ± 0.47 seconds        | 0.150 ± 0.0030 seconds      |
> | 10                        | 18.33 ± 2.13 seconds                             | 6.30 ± 0.43 seconds        | 0.159 ± 0.0034 seconds      |
> | 15                        | 14.19 ± 0.41 seconds                             | 4.30 ± 1.72 seconds        | 0.157 ± 0.0033 seconds      |
> | 20                        | 14.36 ± 0.47 seconds                             | 6.13 ± 0.62 seconds        | 0.148 ± 0.0027 seconds      |
> | 25                        | 14.86 ± 0.11 seconds                             | 7.17 ± 0.46 seconds        | 0.158 ± 0.0022 seconds      |
> |                           |                                                  | **SV‑CMA‑ES with RBF kernel** |                          |
> | 5                         | 10.13 ± 0.21 seconds                             | 6.01 ± 0.23 seconds        | 0.353 ± 0.0060 seconds      |
> | 10                        | 10.34 ± 0.10 seconds                             | 6.28 ± 0.13 seconds        | 0.359 ± 0.0064 seconds      |
> | 15                        | 11.55 ± 0.62 seconds                             | 6.30 ± 0.85 seconds        | 0.360 ± 0.0049 seconds      |
> | 20                        | 12.08 ± 0.10 seconds                             | 6.90 ± 0.19 seconds        | 0.371 ± 0.0021 seconds      |
> | 25                        | 12.77 ± 0.22 seconds                             | 6.90 ± 0.94 seconds        | 0.378 ± 0.0102 seconds      |

---

### Official Review · Reviewer_GvTY · 2025-07-03

**Clarity:** 3
**Significance:** 3
**Originality:** 2
**Rating:** 4
**Confidence:** 4

**Summary:**

Efficient robotic exploration requires information-aware coverage of the domain, avoiding well-understood areas, and focuses more attention on areas that are not well-understood. The paper proposes two methods to diversify Stein Variational Ergodic Search (SVES), which reduce ergodic cost and produce richer trajectory sets than SVES.

**Questions:**

1.	Please provide runtime analysis on a common hardware platform. Ideally benchmark with different number of particles, and problem dimensions.
2.	Are there other efficient popular heuristics other than the median heuristic for selecting bandwidth of kernels? Might be interesting to benchmark these.

**Ethical Concerns:**

["NO or VERY MINOR ethics concerns only"]

**Limitations:**

yes

**Quality:**

2

**Strengths And Weaknesses:**

The paper does a good job of explaining the problem statement, as well as explaining the underlying SVES and Signature Transform and Signature Kernel. Qualitatively, the signature kernel seems to produce richer and diverse trajectories than the other methods in Figure 2. This is further reinforced in Figure 3, where the trajectory diversity is significantly higher than the other methods, while the trajectory and ergodic costs are marginally lower. The signature kernel really stands out from methods in real-world drone tests. SV-CMA-ES shows similar costs compared to SVGD, while having higher average trajectory diversity.

The authors are honest about the limitations of their method, including dependence on hardware, and scalability of the approach. The one critique of the paper is that it claims SV-CMA-ES is a real-time method (in Contributions subsection), due to elimination of gradient computation, but no analysis is provided on this. I would have liked to see a runtime analysis of SV-CMA-ES versus other methods. The time complexity gives an idea of runtime for the signature kernel, but more quantitative evaluation is required. For the most part however, this is a complete work with supplementary results as well.

Submission is clearly written and well-organized. It seems reproducible as experimental and implementation details are provided. Results seem impactful for more efficient robotic exploration, particularly for active perception, where the robot is trying to map its environment in an unknown setting. Due to real-world tests on a drone, it seems like the method is viable for robotics practitioners, although little analysis of runtime is provided. This method improves prior works, and the SV-CMA-ES method is of considerable interest due to its ability to possibly run real-time on a drone.
The work is original as it tackles the problem of diversifying SVES as well as incorporating gradient-free approaches. It is also clear how the method advances previous works, showcasing both qualitative and quantitative comparisons. The work also provides new insights into the meaning of “diversity” for trajectories and improves upon different metrics, such as ergodic cost, trajectory cost and diversity. However, the motivation of integrating SV-CMA-ES is not well-formulated as the paper says “more efficient search steps”, but no quantitative evaluation is provided to back up this claim.

---

> ### Author Rebuttal · Authors · 2025-07-30
>
> # 1. Please provide runtime analysis on a common hardware platform. Ideally benchmark with different number of particles, and problem dimensions?
>
> We thank the reviewer for requesting a detailed runtime comparison. In the following, we report mean runtimes (over 10 independent runs) for SVGD and SV‑CMA‑ES on a common hardware platform, and discuss implementation caveats.
>
> ## Experimental Setup
> The SVGD implementation was fully JIT‑compiled, whereas SV‑CMA‑ES was only partially JIT‑compiled. No optimizations were applied to the signature kernel, so absolute timings may differ under a more optimized implementation. All experiments were run using the following hardware:
>
> 1. **CPU**: AMD Ryzen 7 7800X3D (8 cores @ 4.2 GHz)
> 2. **Memory**: 32 GB (@ 5600 MT/s)
> 3.  **GPU**: NVIDIA Geforce RTX 5060 Ti
>
>
> ## Results
> | Experiments               | Multiscale constrained exploration (Section 7.1) | 3D Coverage (Section 7.2) | SV‑MPC (Section 7.3)         |
> |---------------------------|--------------------------------------------------|----------------------------|-----------------------------|
> | No. of iterations         | 3000                                             | 1500                       | 100                         |
> | Problem dimensions | $T=100, \ d=2$                            |  $T=50, \ d=3$           | $T=20, \ d=2$ |
> | N particles               |                                                  |                            |                             |
> |                           |                                                  | **SVGD with RBF kernel**   |                             |
> | 5                         | 2.83 ± 0.08 seconds                              | 1.45 ± 0.17 seconds        | 0.092 ± 0.0034 seconds      |
> | 10                        | 2.82 ± 0.12 seconds                              | 1.45 ± 0.11 seconds        | 0.093 ± 0.0034 seconds      |
> | 15                        | 2.90 ± 0.14 seconds                              | 1.40 ± 0.09 seconds        | 0.095 ± 0.0078 seconds      |
> | 20                        | 3.11 ± 0.17 seconds                              | 1.46 ± 0.10 seconds        | 0.090 ± 0.0043 seconds      |
> | 25                        | 2.94 ± 0.15 seconds                              | 1.44 ± 0.13 seconds        | 0.094 ± 0.0055 seconds      |
> |                           |                                                  | **SVGD with Signature kernel** |                          |
> | 5                         | 13.83 ± 0.84 seconds                             | 6.32 ± 0.47 seconds        | 0.150 ± 0.0030 seconds      |
> | 10                        | 18.33 ± 2.13 seconds                             | 6.30 ± 0.43 seconds        | 0.159 ± 0.0034 seconds      |
> | 15                        | 14.19 ± 0.41 seconds                             | 4.30 ± 1.72 seconds        | 0.157 ± 0.0033 seconds      |
> | 20                        | 14.36 ± 0.47 seconds                             | 6.13 ± 0.62 seconds        | 0.148 ± 0.0027 seconds      |
> | 25                        | 14.86 ± 0.11 seconds                             | 7.17 ± 0.46 seconds        | 0.158 ± 0.0022 seconds      |
> |                           |                                                  | **SV‑CMA‑ES with RBF kernel** |                          |
> | 5                         | 10.13 ± 0.21 seconds                             | 6.01 ± 0.23 seconds        | 0.353 ± 0.0060 seconds      |
> | 10                        | 10.34 ± 0.10 seconds                             | 6.28 ± 0.13 seconds        | 0.359 ± 0.0064 seconds      |
> | 15                        | 11.55 ± 0.62 seconds                             | 6.30 ± 0.85 seconds        | 0.360 ± 0.0049 seconds      |
> | 20                        | 12.08 ± 0.10 seconds                             | 6.90 ± 0.19 seconds        | 0.371 ± 0.0021 seconds      |
> | 25                        | 12.77 ± 0.22 seconds                             | 6.90 ± 0.94 seconds        | 0.378 ± 0.0102 seconds      |
>
> ## SVGD vs SV-CMA-ES
> The original Stein Variational Gradient Descent (SVGD) paper (see reference [10] from our paper) identifies the gradient of the log‑density as its primary computational bottleneck. Moreover, comparing the costs of SVGD and SV‑CMA‑ES is nontrivial, since both methods require evaluating gradients of the kernel and of the log‑density at each update. The key difference lies in the cost of obtaining the log‑density gradient itself. When that gradient is easy to compute or available analytically, SVGD will generally be the more efficient choice. By contrast, SV‑CMA‑ES truly shines when the log‑density is nondifferentiable or expensive to evaluate.
>
> In practice, computing the log‑density gradient typically entails two passes through the density function (a forward and a backward pass). Evolution‐strategy (ES) updates, however, are highly parallelizable on GPUs. The ES updates require multiple parallel forward passes of the log density, and computing the weighted recombination step for each particle can all be batched on modern hardware (with say Cholesky decomposition). Thus, even though SV‑CMA‑ES performs more sampling “work,” its overhead beyond objective evaluations is often quite small. If the log density is non‑differentiable or costly to backpropagate, SV‑CMA‑ES can actually deliver faster wall‑clock convergence by avoiding gradient calls altogether.  Moreover SV-CMA-ES can have faster convergence rates when compared to SVGD with well-tuned hyperparameter (see reference [25] from our paper).
>
> # 2. Are there other efficient popular heuristics other than the median heuristic for selecting bandwidth of kernels? Might be interesting to benchmark these.
>
> We appreciate the reviewers request to benchmark the different heuristics and an ablation study on the choice of heuristics will be included in the final revision of the paper. The reviewer may refer to the response provided to the 4th question of reviewer KSg3 for additional information beyond the heuristic.
>
> We found that the median heuristic is an extremely popular choice for choosing the bandwidth across multiple domains and provides some strong guarantees [1]. Some of it's advantages include:
> 1. **Fully data‑driven and tuning‑free**: It requires no extra cross‑validation or held‑out data: you just compute all pairwise distances and take their median.
> 2. **Asymptotic consistency and normality**: Under very mild assumptions (e.g. a mix of two distributions), the empirical median of squared distances converges to the true median of a well‑defined “mixture” distance distribution, and indeed satisfies a central‐limit theorem.
> 3. **Separation‑aware behavior**: If the two underlying distributions are well separated, there’s even a high‑probability gap between within‑ and between‑group distances. In that case, the median will reliably fall among the “within‑group” distances, yielding a bandwidth tuned to the scale of each cluster rather than to the far‑apart intercluster distances.
> 4. **Keeps the kernel “in the sweet spot”:** By construction, it avoids two degenerate extremes; too small a bandwidth or too big a bandwidth. The median sits roughly in the “middle” of the distance spectrum, so you capture meaningful variation without blowing up or collapsing the Gram matrix.
>
>
> Additionally, we employed the median heuristic for bandwidth selection because it is straightforward, computationally
> light, and requires no manual tuning, making our method more accessible to users.
>
> A computationally light heuristic is extremely critical, as it has to be used in very iteration of SVGD or SV-CMA-ES. Although several computationally cheap heuristics (like the mean heuristic, Silverman’s rule‑of‑thumb, and Scott’s rule; see the table below) are available they do not in general enjoy non‑asymptotic or distribution‑free theoretical guarantees.
>
> | Heuristic                                  | Computational Cost     | Manual Tuning |
> |--------------------------------------------|------------------------|---------------|
> | Mean heuristic                             | Low                    | Low           |
> | Silverman’s rule‑of‑thumb                  | Low                    | Low           |
> | Scott’s rule                               | Low                    | Low           |
> | Median heuristic                           | Low to Moderate        | Low           |
> | Quantile‑based heuristic                   | Moderate               | Moderate      |
> | Sheather–Jones direct plug‑in              | Moderate               | Moderate      |
> | k‑nearest‑neighbor bandwidths              | Moderate               | Moderate      |
> | Penalized Comparison to Overfitting (PCO)  | Moderate               | High          |
> | Least‑squares cross‑validation (LSCV)      | High                   | High          |
> | Likelihood cross‑validation                 | High                   | High          |
> | Partitioned cross‑validation               | High                   | High          |
>
> # References
> [1] Damien Garreau, Wittawat Jitkrittum, and Motonobu Kanagawa. Large sample analysis of the median
> heuristic. arXiv preprint arXiv:1707.07269, 2018. https://arxiv.org/abs/1707.07269

---

> > ### Comment · Reviewer_GvTY · 2025-08-03
> >
> > THx for the rebuttal and the data for the mean runtimes.

---

> > ### Comment · Reviewer_GvTY · 2025-08-08
> >
> > Thx for the additional results and more discussion on popular heuristics.

---

### Official Review · Reviewer_KSg3 · 2025-07-03

**Clarity:** 2
**Significance:** 2
**Originality:** 2
**Rating:** 4
**Confidence:** 3

**Summary:**

This paper improves Stein Variational Ergodic Search (SVES) for robotic exploration by making trajectories more diverse and lowering ergodic cost. It does this by using a signature kernel and a gradient-free evolution strategy (SV-CMA-ES). Experiments on various tasks demonstrate these improvements.

**Questions:**

1. Could you expand the related work to compare with other ergodic planners or diversity-focused methods outside SVES?
2. Could comparisons be made to other non-SVES baselines?
3. Is there any hardware or real robot demo?
4. Could you provide more details on how sensitive results are to kernel parameters?

**Ethical Concerns:**

["NO or VERY MINOR ethics concerns only"]

**Final Justification:**

The authors address my questions reasonably. However, my my rating remains the same.

**Limitations:**

Yes, the authors acknowledge key technical limitations, including hardware scalability, computational cost for large particle sets, and practical issues like the signature kernel pushing trajectories into obstacles.

**Paper Formatting Concerns:**

No concerns.

**Quality:**

3

**Strengths And Weaknesses:**

Strengths:
1. Clear technical improvement on SVES by combining signature kernels and SV-CMA-ES.
2. Experiments support claims with consistent gains in trajectory diversity and lower ergodic cost.
3. Well-structured and clearly presented paper.

Weaknesses:
1. Could include more related work to better position the contribution.
2. The main novelty seems to be the combination of known components, not new core methods.
3. Missing baselines from recent literature for broader comparison.

---

> ### Author Rebuttal · Authors · 2025-07-30
>
> # 1. Could you expand the related work to compare with other ergodic planners or diversity-focused methods outside SVES?
>
> We thank the reviewer for pointing out the need for including more related work to better position the contribution.
>
> Classical ergodic planners focus on steering a single trajectory (or a centrally controlled fleet) to match a target spatial distribution (Section 1 and Appendix A). In contrast, our SVES‐diversification methods generate a set of distinct or diverse ergodic trajectories in parallel, leveraging repulsive interactions or signature‐based seeding to ensure both high coverage and trajectory diversity in a unified optimization.
>
> Outside of the ergodic literature, the community has developed a rich variety of quality‐diversity and diversity‐maximization techniques such as MAP-Elites [1], Novelty Search [2] and Determinantal point processes [3]. While these methods successfully assemble static repertoires of diverse solutions, they do not directly optimize for ergodic coverage over time.  Our contribution is to bring diversity‐aware techniques into the ergodic planning domain.
>
> At a high level, MAP‑Elites keeps the single best example for each kind of behavior so you end up with a set of good but different options. Novelty Search a evolutionary-computation paradigm, abandons traditional fitness objectives, instead it rewards behavioral divergence; it rewards experiments that lead to brand‑new behaviors, so the search naturally promoting exploration.  Determinantal point processes (DPP) give a straightforward way to pick a batch of items that are both high‑quality and as different from one another as possible.
>
>
>
> # 2. Could comparisons be made to other non-SVES baselines?
> Prior work suggests that the non-convex form of ergodic search methods are capable of identifying multiple optimal trajectories through variations in initial conditions (see reference [9] from our paper). Hence, the main focus of our work is to produce a set of diverse locally optimal trajectories with effective mode coverage. As long as conditions for optimality are the same across both the SVES based and the non-SVES algorithms both should converge to same or similarly optimal trajectories since SVES guarantees convergence (see [2] from our paper).
>
> # 3. Is there any hardware or real robot demo?
> Thank you for highlighting the importance of real‐world validation. In fact, we have conducted hardware experiments with the Crazyflie drone. Figure 1 in the paper illustrates the paths taken by the Crazyflie drone in a real environment, and the quantitative results are reported in Section 7.2. To further underscore these real‐robot findings, additional images and videos can be added after this rebuttal.
>
> # 4. Could you provide more details on how sensitive results are to kernel parameters?
> We once again appreciate the reviewer suggesting a deeper investigation into bandwidth sensitivity as this is indeed very important and not well presented in the paper.
> We will add the following augments to the Appendix to strengthen the paper.
>
> We employed the median heuristic for bandwidth selection because it is straightforward, computationally light, and requires no manual tuning, making our method more accessible to users. Additionally, refer to [4] for details on the guarantees provided by the median heuristic.
>
> However, we find that setting $h=0.1$ for all kernels in the experiment from Section 7—the signature kernel still outperforms the other kernels. We did not find the optimal bandwidth for each kernel (say by grid searching) and compare them. This limitation is acknowledged in Section 9.
>
> But in theory, we expect the signature kernel to retain its advantage thanks to its theoretical properties (discussed in lines 154-158 \& 768-775 of our paper), most notably, the injectivity of its feature map. For a deeper mathematical treatment, we refer the reviewer to [11,14,15,19,61,62] in our paper. To illustrate one consequence of injectivity, consider three simple paths
> 1. $p_1(t) = (x_1(t),y_1(t)) = (t, 0) \quad t \in [0, 2\pi]\$
>
> 2. $p_2(t) = (x_2(t),y_2(t)) = (t, sin(t)) \quad t \in [0, 2\pi]$
>
> 3. $p_3(t) = (x_3(t),y_3(t)) = (t, sin(t + \pi)) \quad t \in [0, 2\pi]$
>
> It is easy to see that $k^{RBF}(p_1,p_2) = k^{RBF}(p_1,p_3)$. In contrast, because the signature transform is injective on all non–tree‑like paths, no two such trajectories share the same signature, and thus the signature kernel cleanly separates these examples. In general the kernelized DTW's and Global alignment kernel's are not feature maps are not injective as well.
>
> # References
>
> [1] Jean-Baptiste Mouret and Jeff Clune, Illuminating search spaces by mapping elites, 2015,
> arXiv:1504.04909, https://arxiv.org/abs/1504.04909
>
> [2] Joel Lehman and Kenneth O. Stanley. Abandoning objectives: Evolution through the search for novelty
> alone. Evolutionary Computation, 19(2):189–223, Summer 2011. MIT Press, Cambridge, MA, USA.
> https://doi.org/10.1162/EVCO_a_00025
>
> [3] Alex Kulesza. Determinantal Point Processes for Machine Learning. Foundations and Trends® in Ma-
> chine Learning, 5(2–3):123–286, 2012. Now Publishers. http://dx.doi.org/10.1561/2200000044
>
> [4] Damien Garreau, Wittawat Jitkrittum, and Motonobu Kanagawa. Large sample analysis of the median
> heuristic. arXiv preprint arXiv:1707.07269, 2018. https://arxiv.org/abs/1707.07269

---

> > ### Comment · Reviewer_KSg3 · 2025-08-06
> > **Thanks for your rebuttal**
> >
> > Thanks for your rebuttal. The responses seem fair.

---

### Comment · Area_Chair_LKww · 2025-08-03
**Reviewers please respond to the rebuttal!**

Dear reviewers,

if you have not yet responded to the rebuttal of the authors, please do so as soon as possible, since the rebuttal window closes soon.

Please check whether all your concerns have been addressed!  If yes, please consider raising your score.

Best wishes,
your AC

---

### Decision · Program_Chairs · 2025-09-17

**Decision:**

Accept (poster)

**Comment:**

The paper presents ergodic search for robots based on the Signature Kernel Evolution Strategy (SVES) to produce more diverse trajectories.  So they modify an existing methods by plugging in a different kernel (replacing an RBF kernel) and a PDE formulation that allows parallel computation.  The reviewers suggest to improve the paper by adding more evaluation, e.g., not only comparing against other kernel-based methods, but also against kernel-free methods.  Furthermore, evaluation on more environments might be useful.  Nonetheless the paper is well written and the overall opinion of the reviewers is borderline accept, so I suggest an "accept".